# FALCON: Fast Visual Concept Learning by Integrating Images, Linguistic descriptions, and Conceptual Relations

**Lingjie Mei**[*]
MIT CSAIL

**Jiayuan Mao**[*]
MIT CSAIL

**Ziqi Wang**
UIUC

**Chuang Gan**
MIT-IBM Watson AI Lab

**Joshua B. Tenenbaum**
MIT BCS, CBMM, CSAIL

## Abstract

We present a meta-learning framework for learning new visual concepts quickly, from just one or a few examples, guided by multiple naturally occurring data streams: simultaneously looking at images, reading sentences that describe the objects in the scene, and interpreting supplemental sentences that relate the novel concept with other concepts. The learned concepts support downstream applications, such as answering questions by reasoning about unseen images. Our model, namely FALCON, represents individual visual concepts, such as colors and shapes, as axis-aligned boxes in a high-dimensional space (the "*box embedding space*"). Given an input image and its paired sentence, our model first resolves the referential expression in the sentence and associate the novel concept with particular objects in the scene. Next, our model interprets supplemental sentences to relate the novel concept with other known concepts, such as "*X has property Y*" or "*X is a kind of Y*". Finally, it infers an optimal box embedding for the novel concept that jointly 1) maximizes the likelihood of the observed instances in the image, and 2) satisfies the relationships between the novel concepts and the known ones. We demonstrate the effectiveness of our model on both synthetic and real-world datasets.

## 1 Introduction

Humans build a cumulative knowledge repository of visual concepts throughout their lives from a diverse set of inputs: by looking at images, reading sentences that describe the properties of the concept, etc. Importantly, adding a novel concept to the knowledge repository requires only a small amount of data, such as a few images about the concept and a short textual description (Bloom, 2000; Swingley, Daniel, 2010; Carey & Bartlett, 1978). Take Fig. 1 as an example: from just a single image that contains many objects (Fig. 1a), as well as a short descriptive sentence that describes a new concept *red*: "the object left of the yellow cube is red", humans can effortlessly ground the novel word "red" with the visual appearance of the object being referred to (Bloom, 2000). Supplemental sentences such as "red is a kind of color" (Fig. 1b) may provide additional information: "red" objects should be classified based on their hue. This further supports us to generalize the learned concept "red" to objects of various shapes, sizes, and materials. Finally, the acquired concept can be used flexibly in other tasks such as question answering (Fig. 1c).

Our goal is to build machines that can learn concepts that are associated with the physical world in an incremental manner and flexibly use them to answer queries. To learn a new concept, for example, the word *red* in Fig. 1a, the system should 1) interpret the semantics of the descriptive sentence composed of other concepts, such as *left*, *yellow*, and *cube*, 2) instantiate a representation for the novel concept with the visual appearance of the referred object, 3) mediate the concept representation based on the supplemental sentences that describe the property of the concept or relate it with other concepts, and 4) use the learned concept flexibly in different scenarios. A framework that can solve these challenges will allow us to build machines that can better learn from human and communicate with human.

To address these challenges, in this paper, we present a unified framework, FALCON (FAst Learning of novel visual CONcepts). FALCON maintains a collection of embedding vectors for individual

---

[*]Equal contribution.

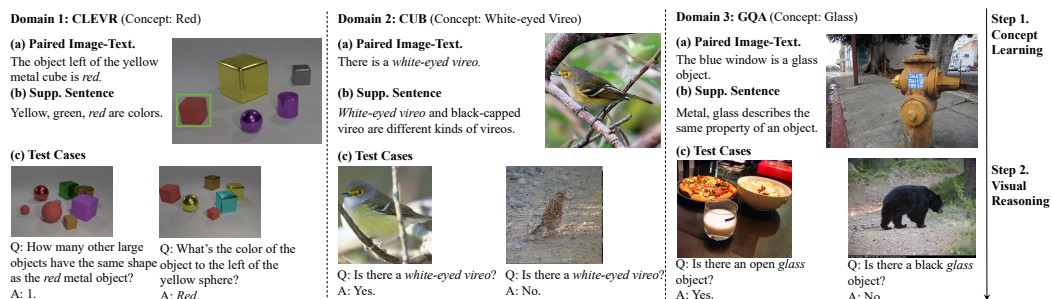

Figure 1: Three illustrative examples of our fast concept learning task. Our model learns from three naturally occurring data streams: (a) looking at images, reading sentences that describe the objects in the scene, and (b) interpreting supplemental sentences that relate the novel concept with other concepts. (c) The acquired novel concepts (e.g., *red* and *white-eyed vireo* transfer to downstream tasks, such as visual reasoning. We present a meta-learning framework to solve this task.

visual concepts, which naturally grows in an incremental way as it learns more concepts. A neuro-symbolic concept learning and reasoning framework learns new concepts by looking at images and reading paired sentences, and use them to answer incoming queries.

Concretely, FALCON represents individual visual concepts, such as colors and shapes, as axis-aligned boxes in a high-dimensional space (the "*box embedding space*" (Vilnis et al., 2018)), while objects in different images will be embedded into the same latent space as points. We say object $X$ has property $Y$ if the embedding vector $X$ is inside the embedding box of $Y$. Given an input image and its paired sentence, our model first resolves the referential expression in the sentence using the previously seen concepts (e.g., *left*, *yellow*, and *cube*) and associate the novel concept with particular objects in the scene. Next, our model interprets supplemental sentences to relate the novel concept with other known concepts (e.g., *yellow*). To infer the box embedding of the novel concept, we train a neural network to predict the optimal box embedding for the novel concept that jointly 1) maximizes the data likelihood of the observed examples, and 2) satisfies the relationships between the novel concepts and the known ones. This module is trained with a meta-learning procedure.

Our paper makes the following contributions. First, we present a unified neuro-symbolic framework for fast visual concept learning from diverse data streams. Second, we introduce a new concept embedding prediction module that learns to integrate visual examples and conceptual relations to infer a novel concept embedding. Finally, we build a protocol for generating meta-learning test cases for evaluating fast visual concept learning, by augmenting existing visual reasoning datasets and knowledge graphs. By evaluation on both synthetic and natural image datasets, we show that our model learns more accurate representations for novel concepts compared with existing baselines for fast concept learning. Systematical studies also show that our model can efficiently use the supplemental concept descriptions to resolve ambiguities in the visual examples. We also provide discussions about the design of different modules in our system.

## 2 RELATED WORKS

**Visual concept learning and visual reasoning.** Visual reasoning aims to reason about object properties and their relationships in given images, usually evaluated as the question-answering accuracy (Johnson et al., 2017a; Hudson & Manning, 2018; Mascharka et al., 2018; Hu et al., 2018). Recently, there has been an increasing amount of work has been focusing on using neuro-symbolic frameworks to bridge visual concept learning and visual reasoning (Yi et al., 2018; Mao et al., 2019; Li et al., 2020). The high-level idea is to disentangle concept learning: association of linguistic units with visual representations, and reasoning: the ability to count objects or make queries. Han et al. (2019) recently shown how jointly learning concepts and metaconcepts can help each other. Our work is an novel approach towards making use of the metaconcepts in a meta-learning setting aiming at boost the learning of novel concepts based on known ones.

**Few-shot visual learning.** Recent work has studied learning to classify visual scene with very limited labeled examples (Vinyals et al., 2016; Sung et al., 2018; Snell et al., 2017) or even without any example (Wang et al., 2018; Kampffmeyer et al., 2019; Tian et al., 2020). For few-shot learning, existing work proposes to compare the similarity, such as cosine similarity (Vinyals et al., 2016) and Euclidean distance (Snell et al., 2017), between examples, while (Sung et al., 2018) introduces a learnable module to predict such similarities. In addition, (Gidaris & Komodakis, 2018) learns a weight generator to predict the classifier for classes with very few examples. (Ravi & Larochelle,

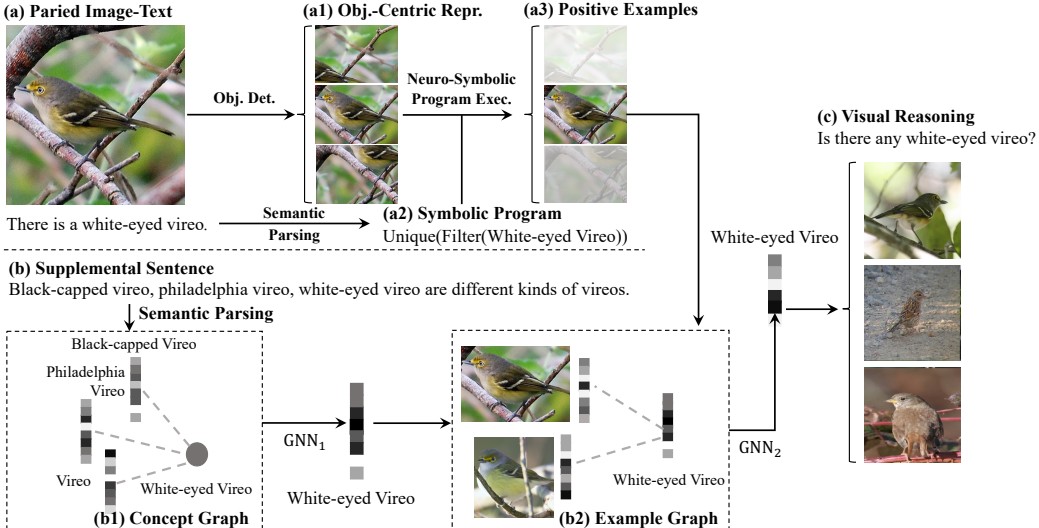

Figure 2: Overview of FALCON-G. FALCON-G starts from extracting object-centric representations and parse input sentences into semantic programs. It executes the program to locate the objects being referred to. The example objects, together with the reconstructed concept graphs are fed into two GNNs sequentially to predict the concept embedding for the "white-eyed vireo". The derived concept embedding can be used in downstream tasks such as visual reasoning.

2017; Finn et al., 2017; Nichol et al., 2018) address this problem by learning the initialization for gradient-based optimization.. (Santoro et al., 2016) used external memory to facilitate learning process, while (Munkhdalai & Yu, 2017) uses meta-knowledge among task for rapid adaptation. Our module design is inspired by these work, but we use a language interface: novel concepts are learnt from paired images and texts and evaluated on visual reasoning tasks.

**Geometric embeddings.** In contrast to representing concepts in vector spaces (Kiros et al., 2014), a geometric embedding framework associates each concept with a geometric entity such as a Gaussian distribution (Vilnis & McCallum, 2015), the intersection of hyperplanes (Vendrov et al., 2016; Vilnis et al., 2018), and a hyperbolic cone (Ganea et al., 2018). Among them, box embeddings (Vilnis et al., 2018) which map each concept to a hyper-box in the high-dimensional space, have been popular for concept representation: (Li, Xiang and Vilnis, Luke and Zhang, Dongxu and Boratko, Michael and McCallum, Andrew, 2019) proposed a smooth training objective, and (Ren et al., 2020) uses box embeddings for reasoning over knowledge graphs. In this paper, we extend the box embedding from knowledge graphs to visual domains, and compare it with other concept embeddings.

## 3   FALCON

The proposed model, FALCON, learns visual concepts by simultaneously looking at images, reading sentences that describe the objects in the scene, and interpreting supplemental sentences that describe the properties of the novel concepts. FALCON learns to learn novel concepts quickly and in a continual manner. We start with a formal definition of our fast and continual concept learning task.

**Problem formulation.** Each concept learning task is a 4-tuple $(c, X_c, D_c, T_c)$. Denote $c$ as the novel concept to be learned (e.g., *red*). Models learn to recognize red objects by looking at paired images $x_i$ and sentences $y_i$: $X_c = \{(x_i, y_i)\}$. Optionally, supplementary sentences $D_c = \{d_i\}$ describe the concept $c$ by relating it to other known concepts. After learning, the model will be tested on downstream tasks. In this paper, we specifically focus on visual reasoning: the ability to answer questions about objects in the testing set $T_c$, which is represented as pairs of images and questions.

There are two possible options to approach this problem. One is manually specifying rules to compute the representation for the new concept. In this paper, we focus on a meta-learning approach: to build a system that can learn to learn new concept. Our training data is data tuples for a set of training concepts (base concepts, $C_{base}$). After training, the system is evaluated on a collection of novel concepts ($C_{test}$). That is, we will provide our system with $X_c$ and $D_c$ for a novel concept $c$, and test it on visual reasoning data $T_c$. Thus, the system works in a continual learning fashion: the description of a new concept depends on a previously learned concept.

**Overview.** Fig. 2 gives an overview of our proposed model, FALCON. Our key idea is to represent each concept as an axis-aligned box in a high-dimensional space (the "box embedding space", Section 3.1). Given example images $x_i$ and descriptions $y_i$ (Fig. 2a), FALCON interprets the referential expression in $y_i$ as a symbolic program (Fig. 2a2). An neuro-symbolic reasoning module executes the inferred program to locate the object being referred to (Fig. 2a3), see Section 3.2. Meanwhile, supplementary descriptions $D_c$ (Fig. 2b) will be translated into relational representations of concepts (Fig. 2b1), i.e., how the new concept $c$ relates to other known concepts. Based on the examples of the novel concept and its relationships with other concepts, we formulate the task of novel concept learning as learning to infer the best concept embedding for $c$ in the box embedding space, evaluated on downstream tasks (Fig. 2c). Once the model learned, we will be using the same neuro-symbolic module for answering questions in the testing set $T_c$.

## 3.1 VISUAL REPRESENTATIONS AND EMBEDDING SPACES

Given the input image, our model extracts an object-based representation for the scene (Fig. 2a1). This is done by first using a pretrained Mask-RCNN (He et al., 2017) to detect objects in the image. The mask for each object, paired with the original image is sent to a ResNet-34 (He et al., 2016) model to extract its visual representation, as a feature vector. For each pair of objects, we will concatenate their feature vectors to form the relational representation between this pair of objects.

The feature vector of each object will be mapped into a geometric embedding space with a fully-connected layer. Each object-based concepts (such as object categories, colors) will be mapped into the *same* embedding space. Similarly, the relational representation of object pairs, as well as relational concepts (such as spatial relationships) will be mapped into another space. Here, we focus on the *box embedding space* (Vilnis et al., 2018) since it naturally models the entailment relationships between concepts. It is also worth noting that other embedding spaces are also compatible with our framework, and we will empirically compare them in Section 4.

**Box embedding.** We slightly extend the box embedding model for knowledge graphs (Vilnis et al., 2018) into visual domains. Throughout this section, we will be using object-based concepts as an example. The idea can be easily applied to relational concepts as well.

Each object-based concept is a tuple of two vectors: $e_c = (\text{Cen}(e_c), \text{Off}(e_c))$, denoting center and offset of the box embedding, both in $\mathbb{R}^d$, where $d$ is the embedding dimension. Intuitively, each concept corresponds to an axis-aligned box (hypercube) in the embedding space, centered at $\text{Cen}(e_c)$ and have edge length $2 \cdot \text{Off}(e_c)$. Each object $o$ is also a box: $(\text{Cen}(e_o), \delta)$, centered at $\text{Cen}(e_o)$ with a fixed edge length of $\delta = 10^{-6}$. We constrain that all boxes should reside within $[-\frac{1}{2}, \frac{1}{2}]^d$.

For each box embedding (either an embedding of an object, or a concept), we define helper functions $\text{Min}(e) = \text{Cen}(e) - \text{Off}(e)$ and $\text{Max}(e) = \text{Cen}(e) + \text{Off}(e)$ to denote the lower- and upper-bound of the a box $e$. The denotational probability is the volume of the box: $\Pr[e] = \prod_i (\text{Max}(c)_i - \text{Min}(c)_i)$, where subscript $i$ denotes the $i$-th dimension of vectors. Obvsiouly, $\Pr[e] \in [0, 1]$. Moreover, we define the joint distribution of two boxes $e_1$ and $e_2$ as the volume of their intersection: $\Pr[e_1 \cap e_2] = \prod_i \max((M(e_1, e_2)_i - m(e_1, e_2)_i), 0)$, where $M(e_1, e_2) = \text{Max}(e_1) \vee \text{Max}(e_2), m(e_1, e_2) = \text{Min}(e_1) \wedge \text{Min}(e_2)$. $\wedge$ and $\vee$ are element-wise max and min operators. Finally, we define the conditional probability $\Pr[e_1|e_2] = \Pr[e_1 \cap e_2] / \Pr[e_2]$. This notation is useful in classifying objects: we will use $\Pr[e_{red}|e_{obj}]$ as the classification confidence of *obj* being *red*. Similarly, it also handles entailment of concepts: $\Pr[e_{c'}|e_c]$ readily denotes the probability that concept $c$ entails concept $c'$. This well suits our need for modeling entailment relations between concepts: in Fig. 2, the supplemental sentence implies "*white-eyed vireo* entails *vireo*". In box embedding, this means: any instance inside the box "white-eyed vireo" should also be inside the box "vireo." We leave detailed discussions and comparisons with other embedding spaces in Appendix Section C.4.

The formulation above has non-continuous gradient at the edges of the box embeddings, following (Li, Xiang and Vilnis, Luke and Zhang, Dongxu and Boratko, Michael and McCallum, Andrew, 2019), we "smooth" the computation by replacing all $\max(\cdot, 0)$ operations with the softplus operator: $\text{softplus}(x) = \tau \log(1 + \exp(x/\tau))$, where $\tau$ is a scalar hyperparameter.

## 3.2 NEURO-SYMBOLIC REASONING

The neuro-symbolic reasoning module processes all linguistic inputs by translating them into a sequence of queries in the concept embedding space, organized by a program layout. Specifically, all natural language inputs, including the sentences $y_i$ paired with the image examples, the supplementary

sentences $d_i$, and questions in $T_c$ will be parsed into symbolic programs, using the same grammar as (Han et al., 2019). Visually grounded questions including $y_i$ and questions in $T_c$ will be parsed into programs with hierarchical layouts. They contain primitive operations such as `Filter` for locating objects with a specific color or shape, and `Query` for query the attribute of a specified object. Appendix Section C contains a formal definition of grammar. Meanwhile, supplementary sentences $d_i$ will be translated as conjunctions of fact tuples: e.g., (red, is, color), which denotes the extra information that the new concept *red* is a kind of *color*. We use pre-trained semantic parsers to translate the natural language inputs into such symbolic representations.

Next, we locate the object being referred to in each $(x_i, y_i)$ pair. This is done by executing the program on the extracted object representations and concept embeddings. Specifically, we apply the neuro-symbolic program executor proposed in (Mao et al., 2019). It contains deterministic functional modules for each primitive operations. The execution result is a distribution $p_j$ over all objects $j$ in image $x_i$, interpreted as the probability that the $j$-th object is being referred to.

The high-level idea of the neuro-symbolic program executor is to smooth the boolean value in deterministic program execution into scores ranging from 0 to 1. For example, in Fig. 2, the execution result of the first `Filter[`*white-eyed vireo*`]` operation yields to a vector $v$, with each entry $v_i$ denoting the probability that the $i$-th object is red. The scores are computed by computing the probability $\Pr[e_{white-eyed-vireo}|e_i]$ for all object embeddings $e_i$ in the box embedding space. The output is fully differentiable w.r.t. the concept embeddings and the object embeddings. Similarly, the concept embeddings can be used to answer downstream visual reasoning questions.

### 3.3 LEARNING TO LEARN CONCEPTS

The embedding prediction module takes the located objects $\{o_i^{(c)}\}$ of the novel concept $c$ in the image, as well as the relationship between $c$ and other concepts as input. It outputs a box embedding for the new concept $e_c$. Here, the relationship between the concepts is represented as a collection of 3-tuples $(c, c', rel)$, where $c'$ is an known concept and $rel$ denotes the relationship between $c$ and $c'$.

At a high level, the inference process has two steps. First, we infer $N$ candidate box embeddings based on their relationship with other concepts, as well as the example objects. Next, we select the concept embedding candidate $e_c$ with the highest data likelihood: $\prod_{i=1}^{M} \Pr[e_c|o_i^{(c)}]$, where $i$ indexes over $M$ example objects. The data likelihood is computed by the denotational probabilities. Following we discuss two candidate models for generating candidate concept embeddings.

**Graph neural networks.** A natural way to model relational structures between concepts and example objects is to use graph neural networks (Hamilton et al., 2017). Specifically, we represent each concept as a node in the graph, whose node embeddings are initialized as the concept embedding $e_c$. For each relation tuple $(c, c', rel)$, we connect $c$ and $c'$. The edge embedding is initialized based on their relationships $rel$. We denote the derived graph as $G_{concept}$ (Fig. 2 (b1)). We also handle example objects in a similar way: for each example object $o_i^{(c)}$, we initialize a node with its visual representation, and connect it with $c$. We denote the derived graph as $G_{example}$ (Fig. 2 (b2)).

To generate candidate concept embeddings, we draw $N$ initial guesses $e_c^{i,0}$, $i = 1, \cdots N$ i.i.d. from the a prior distribution $p_\theta$. For each candidate $e_c^{i,0}$, we feed it to two graph neural networks sequentially:

$$e_c^{i,0} \sim p(\theta), \qquad\qquad e_c^{i,1} = \text{GNN}_1(e_c^{i,0}, G_{concept}),$$

$$e_c^{i,2} = \text{GNN}_2(e_c^{i,1}, G_{example}), \qquad e_c = e_c^{k,2}; k = \arg\max_i \prod_{j=0}^{M} \Pr[e_c^{i,2}|o_i^{(c)}].$$

Note that $\text{GNN}_1$ and $\text{GNN}_2$ are different GNNs. We use the output $e_c$ as the concept embedding for the novel concept $c$. We denote this model as FALCON-G.

**Recurrent neural networks.** A variant of our model, FALCON-R, replaces the graph neural network in FALCON-G with recurrent neural networks. Specifically, we process each edge in the graph in a sequential order. We first sample $e_c^{i,0}$ from $p(\theta)$ and use it to initialize the hidden state of an RNN cell $\text{RNN}_1$. Next, for each edge $(c, c', rel)$ in $G_{concept}$, we concatenate the concept embedding of $c'$ and the edge embedding for $rel$ as the input. They are fed into $\text{RNN}_1$ sequentially. We use the last hidden state of the RNN as $e_c^{i,1}$. We process all edges in $G_{example}$ in the same manner, but with a different RNN cell $\text{RNN}_2$. Since all edges in a graph are unordered, we use random orders for all edges in both graphs. The implementation details for both the GNN and the RNN model can be found in our supplemental material.

**Training objective.** We evaluate the inferred $e_c$ on downstream tasks. Specifically, given an input question and its paired image, we execute the latent program recovered from the question based on the image, and compare it with the groundtruth answer to compute the loss. Note that since the output of the neuro-symbolic program executor is fully differentiable w.r.t. the visual representation and concept embeddings, we can use gradient descent to optimize our model. The overall training objective is comprised of two terms: one for question answering $\mathcal{L}_{QA}$ and the other for the prior distribution matching: $\mathcal{L}_{meta} = \mathcal{L}_{QA} - \lambda_{\text{prior}} \log p_\theta(e_c)$ is the final loss function.

### 3.4 TRAINING AND TESTING PARADIGM

The training and testing pipeline for FALCON consists of three stages, namely the pre-training stage, the meta-training stage, and the meta-testing stage. We process a dataset by splitting all concepts into three groups: $C_{base}$, $C_{val}$, and $C_{test}$, corresponding to the base concepts, validation concepts, and test concepts. Recall that each concept in the dataset is associated with a 4-tuple $(c, X_c, D_c, T_c)$. In pre-training stage, our model is trained on all concepts $c$ in $C_{base}$ using $T_c$, to obtain all concept embeddings for $c \in C_{base}$. This step is the same as the concept learning stage of (Mao et al., 2019), except that we are using a geometric embedding space. In the meta-training stage, we fix the concept embedding for all $c \in C_{base}$, and train the GNN modules (in FALCON-G) or the RNN modules (in FALCON-R) with our meta-learning objective $\mathcal{L}_{meta}$. In this step, we randomly sample task tuples $(c, X_c, D_c, T_c)$ from concepts in $C_{base}$, and use concepts in $C_{val}$ for validation and model selection. In the meta-testing stage, we evaluate models by its downstream task performance on concepts drawn from $C_{test}$. For more details including hyperparameters, please refer to Appendix Section B.

## 4 EXPERIMENTS

### 4.1 DATASET

We procedurally generate meta-training and testing examples. This allows us to have fine-grained control over the dataset distribution. At a high-level, our idea is to generate synthetic descriptive sentences and questions based on the object-level annotations of images and external knowledge bases such as the bird taxonomy. We compare FALCON with end-to-end and other concept-based method on two datasets: CLEVR (Johnson et al., 2017a) and CUB (Wah et al., 2011).

**CUB Dataset.** The CUB-200-2011 dataset (CUB; Wah et al., 2011) contains 12K images of 200 categories of birds. We split these 200 classes into 100 base, 50 validation and 50 test classes following Hilliard et al. (2018). We expand the concept set by including their hypernyms in the bird taxonomy (Sullivan et al., 2009), resulting in 211 base, 83 validation, and 72 test concepts. We consider the two concept relations in CUB: *hypernym* and *cohypernym* (concepts that share the same hypernym). Examples could be found in Fig. 4.

**CLEVR dataset.** We also evaluate our method on the CLEVR dataset (Johnson et al., 2017a). Each image contains 3D shapes placed on a tabletop with various properties – shapes, colors, types of material and sizes. These attribute concepts are grouped into 10 base, 2 validation and 3 test concepts. We create four splits of the CLEVR dataset with different concept split. We generate the referential expressions following (Liu et al., 2019) and question-answer pairs following (Johnson et al., 2017a). Examples could be found in Fig. 4. We consider only one relation between concepts: *hypernym*. For example, *red* is a kind of *color*.

**GQA dataset.** Our last dataset is thee GQA dataset (Hudson & Manning, 2019), which contains image associated with a scene graph annotating the visual concepts in the scene. We select a subset of 105 most frequent concepts. These concepts are grouped into 89 base concepts, 6 validation and 10 test concepts. Examples generated in this dataset can be found in Fig. 4. We consider only one relation between concepts: *hypernym*, e.g. *standing* is a kind of *pose*.

### 4.2 BASELINE

We consider two kinds of approaches: *end-to-end* and *concept-centric*. In *end-to-end* approaches, example images and other descriptive sentences ($X_c$ and $D_c$) are treated as an additional input to the system when responding to each test query in $T_c$. In this sense, the system is not "learning" the novel concept $c$ as an internal representation. Instead, it reasons about images and texts that is related to $c$ and has been seen before, in order to process the queries. Their implementation details can be found in Appendix Section A.

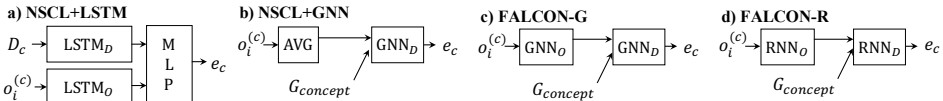

Figure 3: Different models for predicting concept embedding $e_c$. They differ in how they process $D_c$: supplemental sentences, $G_{concept}$: concept graphs, and $o_i^{(c)}$: objects being referred to.

**CNN+LSTM** is the first and simplest baseline, where ResNet-34 (He et al., 2016) and LSTM (Hochreiter & Schmidhuber, 1997) are used to extract image and text representations, respectively. They are fed into a multi-layer perceptron to predict the answer to test questions.

**MAC** (Hudson & Manning, 2018), is a state-of-the-art end-to-end method for visual reasoning. It predicts the answer based on dual attention over images and texts.

We also consider a group of *concept-centric* baselines, clarified in 3. Like FALCON, they all maintain a set of concept embeddings in their memory. The same neuro-symbolic program execution model is used to answer questions in $T_c$. This isolates the inference of novel concept embeddings.

**NSCL+LSTM**, the first baseline in this group uses the same semantic parsing and program execution module to derive the example object embeddings $o_i^{(c)}$ as FALCON. Then, it uses an LSTM to encode all $o_i^{(c)}$ into a single vector, and uses a different LSTM to encode the natural language sentences in $D_c$ into another vector. These two vectors are used to compute the concept embedding $e_c$.

**NSCL+GNN** is similar to NSCL+LSTM except that supplemental sentences $D_c$ are first parsed as conceptual graphs. Next, we initialize the concept embedding in the graph by taking the "average" of visual examples. We then use the same GNN module as FALCON on the conceptual graph $G_{concept}$ to derive $e_c$. Note that NSCL+GNN only use a single GNN corresponding to $G_{concept}$, while FALCON-G uses two GNNs—one for concept graphs and the other for example graphs.

We test concept-centric models on two more embedding spaces besides the box embedding.

**Hyperplane**: object $o$ and concept $c$ correspond to vectors in $\mathbb{R}^d$, denoted as $e_o$ and $e_c$, respectively. We define $\Pr[e_o|e_c] = \sigma(e_o^T e_c/\tau - \gamma)$, where $\sigma$ is the sigmoid function, $\tau = 0.125$, and $\gamma = 2d$.

**Hypercone**: object $o$ and concept $c$ correspond to vectors in $\mathbb{R}^d$, denoted as $e_o$ and $e_c$, respectively. We define $\Pr[e_o|e_c] = \sigma((\langle e_o, e_c \rangle - \gamma)/\tau)$, where $\langle \cdot, \cdot \rangle$ is the cosine similarity function, and $\tau = 0.1, \gamma = 0.2$ are scalar hyperparameters. This embedding space has been proved useful in concept learning literature (Gidaris & Komodakis, 2018; Mao et al., 2019).

### 4.3    RESULTS ON CUB

We evaluate model performance on downstream question-answering pairs in $T_c$ for all test concepts $c \in C_{test}$. There is only one example image for each concept ($|X_c| = 1$). Visual reasoning questions on CUB are simple queries about the bird category in this image. Thus, it has minimal visual reasoning complexity and focuses more on the fast learning of novel concepts.

The results are summarized in Table 1, with qualitative examples in Fig. 4. In general, our model outperforms all baselines, suggesting the effectiveness of our concept embedding prediction. FALCON-R with box embeddings achieves the highest performance. We conjecture that this is because box embeddings can better capture hierarchies and partial order relationships between concepts, such as *hypernyms*. NSCL+LSTM and GNN models have inferior performance with box embeddings, probably because their neural modules cannot handle complex geometric structures like box containment. Unlike them, we use a sample-update-evaluate procedure to predict box embeddings.

By comparing concept-centric models based on the same embedding space (e.g., box embeddings), we see that translating the descriptive sentences $D_c$ into concept graphs significantly improves the performance. Thus, NSCL+LSTM has an inferior performance than other methods. NSCL-GNN and FALCON-G are very similar to each other, except that in NSCL-GNN, the initial embedding of the concept $e_c$ is initialized based on the object examples, while in FALCON-G, we use multiple random samples to initialize the embedding and select the best embedding based on the likelihood function. Moreover, FALCON-G and FALCON-R has similar performance, and it is worth noting that FALCON-R has the advantage that it can perform online learning. That is, edges in $G_{concept}$ and $G_{example}$ of concept $c$ can be gradually introduced to the system to improve the accuracy of the corresponding concept embedding $e_c$.

| Model(Type) | QA Accuracy | | |
|---|---|---|---|
| CNN+LSTM (E2E) | 51.37 | | |
| MAC (E2E) | 73.88 | | |
| | Box | Hyperplane | Hypercone |
| NSCL+LSTM (C.C.) | 79.53 | 71.71 | 72.48 |
| NSCL+GNN (C.C.) | 78.50 | 67.60 | 72.82 |
| FALCON-G (C.C.) | **81.33** | 74.11 | 68.44 |
| FALCON-R (C.C.) | 79.83 | 72.82 | 69.74 |

Table 1: Fast concept learning performance on the CUB dataset. We use "E2E" for end-to-end methods and "C.C." for concept-centric methods. Our model FALCON-G and FALCON-R outperforms all baselines.

| Model(Type) | QA Accuracy | | |
|---|---|---|---|
| CNN+LSTM (E2E) | 51.50 | | |
| MAC (E2E) | 73.55 | | |
| | Box | Hyperplane | Hypercone |
| NSCL+LSTM (C.C.) | 72.23 | 63.18 | 68.48 |
| NSCL+GNN (C.C.) | 73.38 | 62.00 | 72.91 |
| FALCON-G (C.C.) | **76.37** | 64.60 | 70.42 |
| FALCON-R (C.C.) | 75.84 | 63.32 | 69.69 |

Table 2: Fast concept learning with only visual examples, evaluated on the CUB dataset. The task is also referred to as one-shot learning. In this setting, NSCL+GNN will fallback to a Prototypical Network (Snell et al., 2017).

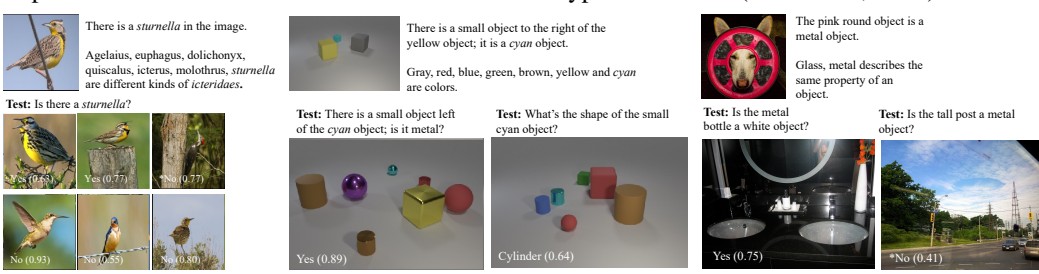

Figure 4: Qualitative results of FALCON-G on the CUB and CLEVR dataset. The bottom left corner of each images shows the model prediction and the confidence score. ∗ marks a wrong prediction.

We summarize our key findings as follows. First, compared with end-to-end neural baselines (CNN+LSTM, MAC), explicit concept representation (NSCL, FALCON) is helpful (Table 3). Second, comparing modules for predicting concept embeddings from supplemental sentences $D_c$: LSTMs over texts $D_c$ (NSCL+LSTM) v.s. GNNs over concept graphs $G_{concept}$ (NSCL+GNN, FALCON-G), we see that, explicitly representing supplemental sentences as graphs and using GNNs to encode the information yields the larger improvement. Third, comparing modules for predicting concept embeddings from visual examples $o_i^{(c)}$: heuristically taking the average (NSCL+GNN) v.s. GNNs on example graphs (FALCON-G), we see that using example graphs is better. These findings will be further supported by results on the other two datasets.

**Case study: few-shot learning.** Few-shot learning (i.e., learning new concepts from just a few examples) is a special case of our fast concept learning task, where $D_c = \varnothing$, i.e., there is no supplemental sentences to relate the novel concept $c$ with other concepts.

Results for this few-shot learning setting are summarized in Table 2. Our model FALCON outperforms other baselines with a significant margin, showing the effectiveness of our sample-update-select procedure for inferring concept embeddings. In this setting, NSCL+GNN computes the novel concept embedding solely based on the "average" of visual examples, and thus being a Prototypical Network (Snell et al., 2017). Note that it cannot work with the box embedding space since the offset of the box cannot be inferred from just one input example.

**Extension: continual learning.** Another advantage of concept-centric models is that it supports the cumulative growth of the learned concept set. That is, newly learned concepts during the "meta-testing" stage can serve as supporting concepts in future concept learning. This is not feasible for end-to-end approaches as they require all concepts referred to in the supplemental sentence in the base concept set $C_{base}$. We extend concept-centric approaches to this continual learning setting, where test concepts $C_{test}$ are learned incrementally, and new concepts might relate to another concept in $C_{test}$ that has been learned before. In this setup, FALCON-G achieves the highest accuracy (79.67%), outperforming the strongest baseline NSCL+LSTM by 4.46%. Details are in Appendix Section C.1.

**Ablation: the number of base concepts.** We examine how the number of base concepts contributes to the model performance. We design a new split of the CUB dataset containing 50 training species (130 base concepts), compared to 100 training species (211 base concepts) of the original split. Our FALCON-G model's accuracy drops from 81.33% to 76.32% when given fewer base concepts. This demonstrates that transfering knowledge from concepts already learned is helpful. See Appendix D.1.

**Ablation: the number of related concepts in supplemental sentences.** We provide an ablation

| Model | Example Only | Example+Supp. | $\Delta$ |
|---|---|---|---|
| CNN+LSTM | 41.92 | 42.69 | 0.77 |
| MAC | 61.81 | 62.05 | 0.24 |
| NSCL+LSTM | 56.82 | 61.35 | 4.53 |
| NSCL+GNN | 65.01 | 84.11 | 19.11 |
| FALCON-G | 68.47 | **88.40** | 19.93 |
| FALCON-R | 68.57 | 87.36 | 18.78 |

Table 3: Fast concept learning performance on the CLEVR dataset. We compare both Example only and the Example +Supp. settings. The results are evaluated and averaged on the four splits of the dataset. See the main text for analysis.

on the effects of the number of related concepts in supplemental sentences. We use supplemental sentences containing 0%(no supp. sentence), 25%, 50%, 75%, and 100%(the original setting) of all related concepts. Our model FALCON-G yields an accuracy of 76.37%, 80.20%, 80.78%, 81.20%, 81.33%, respectively. This result shows that more related concepts included in supplemental sentences lead to higher accuracy. Details are in Appendix D.2.

## 4.4 Results on CLEVR

We use a similar data generation protocol for the CLEVR dataset as CUB. While CLEVR contains concepts with simpler appearance variants, it has a more complicated linguistic interface. More importantly, all concepts in CLEVR are disentangled and thus allow us to produce compositional object appearances. We will exploit this feature to conduct a systematical study on concept learning from biased data. Our framework FALCON provides a principled solution to learning from biased data, which has been previously studied in (Geirhos et al., 2019) and (Han et al., 2019).

Our main results are summarized in Table 3. Fig. 4 provides qualitative examples. All concept-centric models use the box embedding space. We also compare model performance on the "few-shot" learning setting, where no supplemental sentences are provided. Due to the compositional nature of CLEVR objects, learning novel concepts from just a single example is hard, and almost always ambiguous: the model can choose to classify the examples based on either its shape, or color, or material. As a result, all models have a relatively poor performance when there are no supplementary sentences specifying the kind of novel concept (color/shape/etc.).

**Case study: learning from biased data.** We provide a systematical analysis on concept learning from biased data. In this setup, each test concept $c$ is associated with another test concept $c'$ (both unseen during training). We impose the constraint that all example objects of concept $c$ in $X_c$ also has concept $c'$. Thus, $c$ and $c'$ is theoretically ambiguous given only image examples. The ambiguity can be resolved by relating the concept $c$ with other known concepts. Overall, our model, FALCON-G achieves the highest accuracy (88.29%) in this setting (MAC: 61.93%, NSCL+GNN 84.83%). Moreover, our model shows the most significant performance improvement after reading supplemental sentences. Details can be found in Appendix Section C.2.

## 4.5 Results on GQA

Following a similar data generation protocol as the previous datasets, we generate questions on the GQA dataset that ask whether a linguistically grounded object in the picture is associated with a novel concept. Since each GQA picture contains a large number of concepts in different image regions, it provides a more complex linguistic interface than the CUB dataset and more diverse visual appearances than the CLEVR dataset. We exploit this advantage to test our model's performance on real-world images and questions in the GQA dataset.

Our main results are shown in Table 11a in the Appendix Section C.3. Fig. 4 provides qualitative examples. Overall, our model achieves the highest accuracy (55.96%) in the setting with supplementary sentences (MAC: 54.99%, NSCL+LSTM: 55.41%). Due to the visual complexity in the GQA dataset, the exact decision boundary of a concept is hard to determine. Details are in Appendix Section C.3.

## 5 Conclusion

We have presented FALCON, a meta-learning framework for fast learning of visual concepts. Our model learns from multiple naturally occurring data streams: simultaneously looking at images, reading sentences that describe the image, and interpreting supplemental sentences about relationships between concepts. Our high-level idea is to incorporate neuro-symbolic learning frameworks to interpret natural language and associate visual representations with latent concept embeddings. We improve the learning efficiency by meta-learning concept embedding inference modules that integrates visual and textual information. The acquired concepts directly transfer to other downstream tasks, evaluated by the visual question answering accuracy in this paper.

**Acknowledgements.** This work is in part supported by ONR MURI N00014-16-1-2007, the Center for Brain, Minds, and Machines (CBMM, funded by NSF STC award CCF-1231216), the MIT Quest for Intelligence, MIT–IBM AI Lab. Any opinions, findings, and conclusions or recommendations expressed in this material are those of the authors and do not necessarily reflect the views of our sponsors.

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

The appendix is organized as the following. First, in Section A, we present our model details, especially the computation performed by our GNN- and RNN-based model for concept graphs and example graphs. Next, in Section B, we formally defines the training objective for different stages in FALCON. Finally, in Section C we supplement details and examples of our datasets and experimental setups.

# A MODEL DETAILS

## A.1 REMARKS ON THE OVERALL MODEL STRUCTURE

FALCON follows a modularized, concept-centric design pattern: the system maintains a set of concept embeddings, which naturally grows as it learns new concepts. We use separate modules for computing novel concept embeddings (Section 3.3) and processing queries (Section 3.2). These modules directly compute the novel concept embeddings, essentially classifiers on objects and their relationships, and use them to answer complex linguistic queries.

Here, all images and texts processed by the system will be "compressed" as these concept embeddings in a latent space. There are other possible designs, such as storing all training images and texts directly, and re-processing all training data for every single query in $T_c$. Despite being inefficient, as we will show in experiments, such design also shows inferior performance compared with FALCON.

## A.2 CONCEPT EMBEDDING

Throughout the paper, We use a hidden dimension of $d = 100$ for the box embedding space (note that the vector representation for this box embedding space is 200-d since we have both centers and offsets), and $d = 512$ for the hyperplane and the hypercone embedding space. For the box embedding space, we set the smoothing hyperparameter to $\tau = 0.2$, following Li, Xiang and Vilnis, Luke and Zhang, Dongxu and Boratko, Michael and McCallum, Andrew (2019).

## A.3 PRIOR DISTRIBUTION

In both FALCON-G and FALCON-R, we assume a prior $p_\theta$ on concept embeddings, from which we sample the initial guess $e_c^{i,0}$. For box embedding space, we model the prior distribution of $e_c$ so that its center and offset along all $d$ dimensions can be generated by the same Dirichlet distribution parameterized by $\theta$. In particular, we sample $d$ independent samples $(x_{0p}, x_{1p}, x_{2p})_{p \in \{0,1,\cdots d-1\}}$ from a Dirichlet distribution parameterized by $\theta = (\theta_0, \theta_1, \theta_2)$. Then an initial guess for concept embedding of $c$ is given as $\mathrm{Cen}(c)_p = \frac{1}{2}(x_{1p} - x_{0p})$, and $\mathrm{Off}(c)_p = \frac{1}{2}x_{2p}$. For hyperplane and hypercone embedding space, we model the prior as the same Gaussian distribution on all $d$ dimensions, namely $e_c \sim \mathcal{N}(\mathbf{0}, \theta_0 \mathbf{I}_d)$, where $\mathbf{I}$ is the identity matrix and $\theta_0$ a scalar parameter. In both embedding space choices, $\theta$ can be learnt from back-propagation from $\mathcal{L}_{\mathrm{meta}} = \mathcal{L}_{QA} - \lambda_{\mathrm{prior}} \log p_\theta(e_c)$, and we set the hyperparameter $\lambda_{\mathrm{prior}} = 1$.

## A.4 FALCON-G

In FALCON-G, there two graphical neural networks, $\mathrm{GNN}_1$ and $\mathrm{GNN}_2$, which operates on the concept graph $G_{concept}$ and example graph $G_{example}$, respectively. Both of them are composed by stacking two graphical neural network (GNN) layers.

For convenience, we denote the input embedding of node $i$ at layer $\ell \in \{0, 1\}$ as $h_i^{(\ell)}$. At layer $\ell$, we first generate the message from node $j$ to its neighbour $i$ by:

$$m_{ji} = \mathrm{MP}_k(h_j^{(\ell)}, h_i^{(\ell)}),$$

where $k$ denote the relation between $j$ and $i$, where MP is a message passing function, which we will formally define later.

Next, for each node $i$, we aggregate the message passed to $i$ by computing an element-wise maximum (i.e., max pooling) of all neighbouring $j$'s as $h_{\mathcal{N}_G(i)}^{(\ell)}$, where $\mathcal{N}_G(i)$ is the set of all neighboring nodes.

Finally, we update node embeddings for $i$ based on its original embedding $h_i^{(\ell)}$ and aggregated messages $h_{\mathcal{N}_G(i)}^{(\ell)}$ for node $i$. Mathematically,

$$h_i^{(\ell+1)} = \text{UPDATE}(h_i^{(\ell)}, h_{\mathcal{N}_G(i)}^{(\ell)}).$$

Concretely, we implement the message passing model $\text{MP}_k$ based on specific concept embedding spaces. For box embedding spaces, we first split $h_j^{(\ell)}$ as two vectors, denoted as $\text{Cen}(h_j^{(\ell)})$ and $\text{Off}(h_j^{(\ell)})$, respectively. Similarly, we split $h_i^{(\ell)}$ into $\text{Cen}(h_i^{(\ell)})$ and $\text{Off}(h_i^{(\ell)})$. For each hidden dimension $p \in \{0, 1, 2, \cdots, d-1\}$, we concatenate $\text{Cen}_p(h_j^{(\ell)}), \text{Off}_p(h_j^{(\ell)}), \text{Cen}_p(h_i^{(\ell)})$, and $\text{Off}_p(h_i^{(\ell)})$ as a 4-d vector, and applies a two-layer perceptron (MLP) on it. This MLP is shared across all hidden dimensions $p$. For different binary relations $k$, we use different MLPs. In our experiment, the aforementioned MLP has an output dimension of 10 and a hidden dimension of 20.

For the hyperplane and the hypercone embeddings, we simply concatenate $h_j^{(\ell)}$ and $h_i^{(\ell)}$, and send it into a MLP associated with relation $k$. The MLP has 1024 hidden units and 512 output units.

The update model UPDATE is implemented in a similar manner. For the box embedding space, we concatenate the hidden vector associated with each hidden dimension $p$ of $h_i^{(\ell)}$ and $h_{\mathcal{N}_G(i)}^{(\ell)}$ as a 12-d vector. We send it to a feed-forward neural network with 20 hidden units and 4 output units, which is shared across $d$ dimensions. If we denote the four output units as $o_{ip}^{(\ell)}, \delta_{ip}^{(\ell)}, o_{ip}'^{(\ell)}, \delta_{ip}'^{(\ell)}$, we use them to calculate the new box embedding $h_i^{(l+1)}$ as

$$
\begin{aligned}
\text{Cen}(h_{ip}^{(\ell+1)}) &= \text{Cen}(h_{ip}^{(\ell)}) + \sigma(o_{ip}^{(\ell)})\delta_{ip}^{(\ell)}; \\
\text{Off}(h_{ip}^{(\ell+1)}) &= \text{Off}(h_{ip}^{(\ell)}) + \sigma(o_{ip}'^{(\ell)})\delta_{ip}'^{(\ell)},
\end{aligned}
$$

where $\sigma$ is the sigmoid function. For hyperplane and hypercone embeddings spaces, we implement UPDATE by concatenate $h_i^{(\ell)}$ and $h_{\mathcal{N}_G(i)}^{(\ell)}$ and send it to a feed-forward neural network with 1024 hidden units and 1024 output units. If we denote its output as $o_{ip}^{(\ell)}, \delta_{ip}^{(\ell)}$, each having a length of 512, then the update model can be written as

$$h_i^{(\ell+1)} = h_i^{(\ell)} + \sigma(o_i) \odot \delta_i,$$

where $\odot$ denotes element-wise multiplication.

## A.5 FALCON-R

In FALCON-R, concepts graph and example graphs are handled jointly in a unified framework. For each each object example $o_i^{(c)}$ associated with $c$, we treat their relationship as a "concept-example" relation. Thus, we can combine two graphs into a single graph. We denote the resulting graph as $G^*$.

We flatten the graph $G^*$ by randomly order all relational tuples $(c, c', rel)$ in $G^*$. We then feed them sequentially into an RNN cell. We implement the RNN cells $\text{RNN}_1$ and $\text{RNN}_2$ as a GNN used in FALCON-G, but is applied to a 2-node graph.

```
1  for j, i, k in G.edges:
2      split = torch.stack([
3          *h[j].chunk(2, -1),
4          *h[i].chunk(2, -1)
5      ], dim=-1)
6      m[(j, i)] = mlp_1[k](split)
7  for i in G.nodes:
8      h_N[i] = torch.stack([
9          m[(jj, ii)] for (jj, ii) in m if i == ii
10     ]).max(dim=0).values
11 for i in G.nodes:
```

| Operation | Semantics |
|---|---|
| Scene | Return all objects in the scene. |
| Filter | Filter out the set of objects from input objects that have the specific object-level concept. |
| Relate | Filter out the set of objects that have the specific relational concept with as input object. |
| AERelate | Filter out the set of objects that have the same attribute value as the input object. |
| Intersection | Return the intersection of two object sets. |
| Union | Return the union of two object sets. |
| Query | Query the attribute of the input object. |
| AEQuery | Query if two objects have the same attribute value (e.g., color). |
| Exist | Query if an object set is empty. |
| Count | Return the size of an object set. |
| CountLessThan | Query if the size of the first object set is smaller than the size of the second object set. |
| CountGreaterThan | Query if the size of the first object set is greater than the size of the second object set. |
| CountEqual | Query if the size of the first set is the same as the size of the second object set. |

Table 4: All program operations supported in CLEVR.

```
12    o, delta, o_prime, delta_prime = mlp_2(torch.stack([
13        *h[i].chunk(2, dim=-1),
14        *h_N[i].unbind(dim=-1)
15    ])).unbind(dim=-1)
16    h[i] = torch.cat([
17        center + torch.sigmoid(o) * delta,
18        offset + torch.sigmoid(o_prime) * delta_prime
19    ], dim=-1)
```

Snippet 1: A snippet of one-step concept graph propagation based on the box embedding space. The code is written in PyTorch Paszke et al. (2019). The code snippet updates the node embedding `h[i]` by first computing the messages passing from node `j` to node `i` in the first for-loop, aggregating the messages to node `i` by taking the element-wise max, and finally updating the node embedding of `i`.

### A.6 SEMANTIC PARSER

In semantic parsing module, the sentence is translated into programs of primitive operations that have hierarchical layouts. For CLEVR dataset, the complete list of operation we provided is given in Table 4. For CUB dataset, since there is only one bird in the image, we only provide operations `Scene`, `Filter` and `Exist` listed in 4. Examples of the output of semantic parsing module are provided in Table 5.

Our semantic parser follows the general framework of Seq2seq (Sutskever et al., 2014) and our implementation follows NS-VQA (Yi et al., 2018) which has been developed for VQA tasks. We first tokenize the input sentence into a sequence of word ids. Then, we feed the sequence into a Seq2Seq model and generate a sequence of $(op_i, concept_i)$, where $i = 1, 2, 3, ..., L_{out}$ where $L_{out}$ is the output sequence length. $op_i$'s are operations defined in the DSL, and they form a "flattened" program based on which we will recover a hierarchical program. Some operations such as *filter* has a concept input, in which case we use the corresponding $concept_i$. The semantic parser is trained on 10% of all generated data.

### A.7 NEURO-SYMBOLIC PROGRM EXECUTION

We mostly follow the original design of NSCL as in Mao et al. Mao et al. (2019). The neuro-symbolic executor executes the program based on both the scene representation and the concept embeddings.

| Dataset | Input | Output |
|---|---|---|
| CLEVR (Description Sentences) | The red objects that are both in front of the cyan object and behind the cube is a sphere. | `Filter(Intersection(`
`  Relate(Filter(Scene(),`
`   cyan), front),`
`  Relate(Filter(Scene(),`
`   cube), behind)`
`), red)` |
| CLEVR (Questions) | There is a rubber object to the left of the metal object; What is its color? | `Query(Filter(`
`  Relate(Filter(Scene(),`
`   metal), left),`
`rubber), color)` |
| CUB (Description Sentences) | There is a frigatebird. | `Scene()` |
| CUB (Questions) | Is there a frigatebird? | `Exist(Filter(Scene(),`
`  frigatebird))` |
| GQA (Description Sentences) | The clouds are a white object. | `Filter(Scene(),clouds)` |
| GQA (Questions) | Is the standing man a red object? | `Exist(Filter(Filter(`
`  Filter(Scene(), standing),`
`man), red))` |

Table 5: Example questions and the programs recovered by the semantic parser. For description sentences, the semantic parser parse the referential expression in the sentence into a program. On CUB, since there is only one object in the image, the operation `Scene()` returns the only object in the image.

Lying in the core of execution is the representation of object sets, which are immediate results during the execution. Each set of objects (such as the output of a `Filter` operation, is a vector of length $N$, where $N$ is the number of objects in the scene. Each entry is a score ranging from 0 to 1. We use the same implementation as Mao et al. (2019), with two modifications:

First, the probability of classifying object $o$ as concept $c$, $\Pr[e_o \mid e_c]$ (`ObjClassify` in the NSCL paper), is implemented based on different embedding spaces: box, hypercone and hyperplane. The original NSCL paper uses the hypercone representation.

Second, the probability that two objects having the same attribute (named `AEClassify` in the NSCL paper) are treated as special kinds of relational concepts: `has_same_color`, `has_same_shape`, etc. Thus, the `AERelate` operation is implemented in the same way as `Relate`.

### A.8 BASELINE IMPLEMENTATION

**CNN+LSTM** In CNN+LSTM, we use a ResNet-34 network with a global average pooling layer to extract the image representation. Each image feature is a feature vector of dimension 512. We concatenate the description sentences, supplemental sentences (if any), and test questions with a separator to form the text input. We use a bi-directional LSTM to encode the text into a 512-dimensional vector (we use the output state of the last token). We use a two-layer LSTM, each with a hidden state of dimension 512. The image representation of both images (the example image in $X_c$, and the test image in $T_c$), together with the text encoding, are sent into a multi-layer perceptron (MLP) to predict the final answer. The MLP has two layers, with a hidden dimension of 512.

**MAC** The MAC network takes an image representation (encoded by CNNs) and a text representation (encoded by RNNs) as its input, and directly outputs a distribution over possible answers. Following the original paper Hudson & Manning (2018), we use the first four residual blocks of a ResNet-101 He et al. (2016) to extract image representations, which is a 2D feature map with hidden dimension 512. The image representations of the example image and the test image an stacked channel-wise into a 2D feature map of hidden dimension 1024. The Similar to CNN+LSTM, we

concatenate the description sentences, supplemental sentences (if any), and test questions with a separator to form the text input. For all other parts, we use the same setting as the original paper Hudson & Manning (2018), such as the hidden dimension for all units in MAC.

**NSCL+LSTM**   In NSCL+LSTM, we use the same semantic parsing module and program execution module to derive the example object embeddings $o_i^{(c)}$. These object embeddings are encoded by a bi-directional LSTM into a 512-dimensional vector. We use a separate bi-directional LSTM to encode the $G_{concept}$ into a 512-dimensional vector. Embeddings of related concept $c'$, concatenated with an embedding for the relation $rel$, are fed in to the LSTM one at a time. Two LSTM both use as output the output state associated with the last input, and have two layers and 512 hidden dimension. These output of two LSTMs are concatenated and sent to a linear transform to predict the concept embedding $e_c$. The output of the linear transform $e_c$ will be used by the neuro-symbolic program execution module to predict the final answer to queries. NSCL+LSTM is an ablative model of FALCON-R. It keeps similar designs as FALCON-R except it places visual examples and linguistic guidance on different streams, whereas FALCON-R uses a single recurrent mechanism to capture both the visual and linguistic information.

**NSCL+GNN**   In NSCL+GNN, the supplemental sentence $D_c$ is first parsed into conceptual graph $G_{concept}$. All nodes corresponding to related concepts $c'$ are initialized by the concept embeddings of $c'$. The node corresponding to concept $c$ is initialized from the "average" of all visual examples. For both hyperplane and hypercone embedding spaces, the average is computed by taking the mean of (L2-normalized) example object embeddings. For box embedding spaces, it is computed as the specific boundary (i.e., the minimal bounding box that covers all examples) of visual example embeddings. For meta-learning setting without supplmentary sentences, we use the average box size of all concepts in the training set to initialize the box size for novel concepts, so the NSCL+GNN model would become a special example of the Prototypical Network. The GNN has two layers, and has the same design as our mode FALCON-G. NSCL+GNN is an ablative model of FALCON-G. It keeps all designs of FALCON-G except that it directly computes the "average" of visual examples. In contrast, in FALCON-G, the example object information are gathered with a example graph neural network encoder.

## B   TRAINING DETAILS

Our training paradigm consists of three stages: pre-training, meta-training and meta-testing. Fig. 5 gives an illustrative overview of these three stages. Recall that we have processed the dataset by splitting all concepts into three groups: $C_{base}$, $C_{val}$, and $C_{test}$, and each concept in the dataset is associated with a 4-tuple $(c, X_c, D_c, T_c)$.

In pre-training stage, our model is trained on all concepts $c$ in base concept set $C_{base}$. The concept embedding of $c$ and the visual representations are trained using question-answer pairs from $T_c$. This step is the same as the concept learning stage of Mao et al. (2019), except that we are using a geometric embedding space. In particular, based on the classification confidence scores of object properties and their relations, the model outputs a distribution over possible answers for each recovered program. On the CLEVR dataset, we use the same curriculum learning setting as Mao et al. (2019), based on the number of objects in the image and the complexity of the questions. In the meta-training stage, we fix the concept embedding for all concepts $c \in C_{base}$, and train the GNN modules (in FALCON-G) or the RNN modules (in FALCON-R) with our meta-learning objective $\mathcal{L}_{meta}$. In this step, we randomly sample task tuples $(c, X_c, D_c, T_c)$ of all concepts in $C_{base}$, and use concepts in $C_{val}$ for validation and model selection. In the meta-testing state, we evaluate each model by its downstream task performance on concepts drawn from $C_{test}$.

One issue with end-to-end methods (CNN+LSTM and MAC) is that they assume that the concept vocabulary is known. Upon seeing a new concept, they do not have the mechanism for inducing a new word embedding for this concept $c$. To address this issue, in the meta-training and meta-testing stages, the appearance of the novel concept $c$ in all texts will be replaced by a special token <UNK>.

Our model is trained with the following hyperparameters. During the pre-training stage, the model is trained for 50000 iterations with a batch size of 10, using an Adam optimizer Kingma & Ba (2014) with learning rate 0.0001. During the meta-training stage, the model is trained for 10000

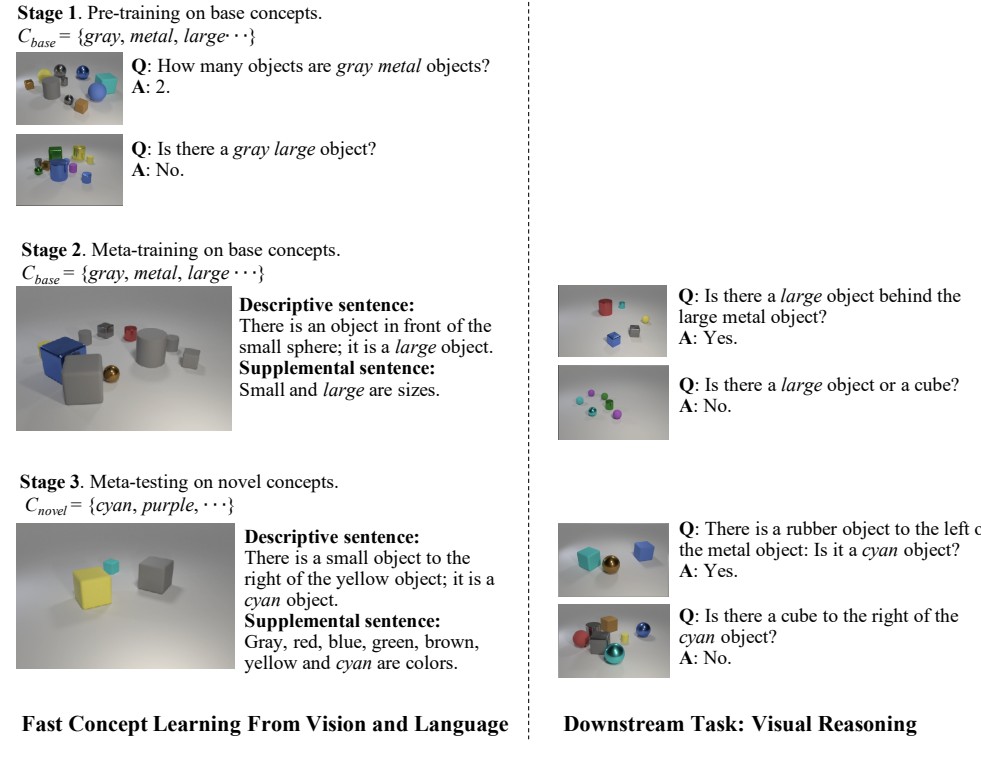

Figure 5: An overview of the overall training paradigm.

iterations with a batch size of 1 (one novel concept per iteration). We train the model with the same optimizer setup (Adam with a learning rate of 0.001). In both stages, we evaluate the model on their corresponding validation set every 1000 iterations. We decrease the learning rate by a factor of 0.1 when the validation accuracy fails to improve for 5 consecutive validation evaluations, i.e. the iteration with the best validation accuracy is more than 5 validations ago.

## C   EXPERIMENT DETAILS

### C.1   THE CUB DATASET

In the CUB dataset, descriptive sentences are generated from the referential expressions produced by the templates in Table 6a, containing only concepts from $\mathcal{C}_{base}$. We provided an approximate of 1.8k descriptive sentences, 1055 of which for $\mathcal{C}_{base}$, 415 for $\mathcal{C}_{val}$ and 360 for $\mathcal{C}_{test}$. Supplemental sentences are generated based on the templates in Table 6b. In supplemental sentences for concept $c \in \mathcal{C}_{base}$, we randomly remove a related concept $c'$ from the sentence with a probability of 0.2. We generate 54k testing questions based on the templates in Table 6c, 31.5k of them paired with concepts in $\mathcal{C}_{train}$, 12.5k of them paired with concepts in $\mathcal{C}_{val}$ and 10k of them paired with concepts in $\mathcal{C}_{test}$. Descriptive sentences, supplemental sentences and testing questions are zipped together to form the input for fast concept learning task. The same template generates Pre-training questions as testing questions. We generated around 11.8k question-image pairs for pre-training. During training, we apply standard data augmentation techniques on the images, including horizontal flip, random resizing and cropping.

**Extension: continual learning.**   In the continual learning setting, concepts that has been learned in the meta-testing instances can be used as supporting concepts for other concepts. We implement this idea as the following. Since CUB is constructed based on a bird taxonomy, we can define the distance between two concepts by the length of their shortest path on the bird taxonomy. We define the *hop* number of each concept as their distance with the closest concepts $c' \in C_{base}$. In the meta-testing

| (a) The descriptive sentence template in CUB. |  |
| --- | --- |
| **Template** | There is a/an `<CONCEPT>`. |
| **Example** | There is a white-eyed vireo. |
| **Description** | `<CONCEPT>` is the word representing concept $c$. |

| (b) The supplemental sentence template in CUB. |  |
| --- | --- |
| **Template** | `<CONCEPT-1>`, `<CONCEPT-2>`$\cdots$ `<CONCEPT-N>` are a kind of `<HYPERNYM>`. |
| **Example** | Black-capped vireo, philadelphia vireo, warbling vireo, white-eyed vireo are a kind of vireos. |
| **Description** | `<CONCEPT-i>` are words representing $c$ and $c'$ that share a *cohypernym* relation with $c$. `<HYPERNYM>` denotes $c$'s hypernym. |

| (c) The test question template in CUB. |  |
| --- | --- |
| **Template** | Is there a/an `<CONCEPT>`. |
| **Example** | Is there a white-eyed vireo. |
| **Description** | `<CONCEPT>` is the word representing concept $c$. |

Table 6: Templates for generating descriptive sentences, supplemental sentences and test questions in CUB.

| Model | Performance | | | Model | Performance | | |
| --- | --- | --- | --- | --- | --- | --- | --- |
|  | Box | Hyperplane | Hypercone |  | Box | Hyperplane | Hypercone |
| NSCL+LSTM | 75.21 | 67.78 | 71.06 | NSCL+LSTM | 71.31 | 62.45 | 67.50 |
| NSCL+GNN | 71.04 | 64.74 | 72.37 | NSCL+GNN | 72.53 | 61.56 | 71.25 |
| FALCON-G | **79.67** | 73.14 | 64.37 | FALCON-G | **75.28** | 64.16 | 68.73 |
| FALCON-R | 78.76 | 65.38 | 68.11 | FALCON-R | 75.01 | 62.78 | 67.96 |

(a) The continual concept learning performance on the CUB dataset.

(b) The continual concept learning performance with only visual examples on the CUB dataset. This task is also referred as few-shot learning, where NSCL+GNN will fall back to a Prototypical Network Snell et al. (2017).

Table 7: The continual concept learning performance on the CUB dataset. Only concept-centric methods can be applied to this setting. FALCON-G and FALCON-R outperform all baselines. The table b serves as comparison with table a. FALCON-G and FALCON-R make the best use supplemental sentences to help concept learning.

stage, we test models with concepts of increasing *hop* numbers. For example, we first teach the model with 1-hop concepts, which directly relates to concepts they have seen in $\mathcal{C}_{base}$. Next, we add 2-hop concepts. During the learning of 2-hop concepts, the concept embedding of those 1-hop concepts are fixed. The detailed result for concept-centric methods for continual concept learning can be found in Table 7a. The accuracy is averaged over all 72 test concepts in $\mathcal{C}_{test}$. For reference, we also show the results of applying these models to the fast concept learning tasks without supplemental sentences in Table 7b (the few-shot learning setting). This results are computed over all 72 concepts in $\mathcal{C}_{test}$, although we do not use any supplemental sentences. Our model makes the best use of supplemental sentences in inducing novel concept embeddings.

## C.2 THE CLEVR DATASET.

CLEVR dataset contains 15 individual concepts, 8 of them associates with colors, 3 associates with shapes, 2 associated with the materials of objects and 2 associated with the sizes of objects. Each split is generated by selecting 2 colors as validation concepts $\mathcal{C}_{val}$, and another 2 colors and 1 shape as testing concepts $\mathcal{C}_{test}$.

(a) The descriptive sentence template in CLEVR.

| | |
|---|---|
| **Template** | There is a `<REFEXP>`; it is `<CONCEPT>`. |
| **Example** | There is a small object to the right of the yellow object; it is a cyan object. |
| **Description** | `<REFEXP>` is a referential expression generated by CLEVR-Ref+ Liu et al. (2019). `<CONCEPT>` is the word for concept $c$. |

(b) The supplemental sentence template in CLEVR.

| | |
|---|---|
| **Template** | `<CONCEPT-1>`,`<CONCEPT-2>`$\cdots$`<CONCEPT-N>` are `<ATTRIBUTE>`. |
| **Example** | Cube, sphere and cylinder are shapes. |
| **Description** | `<CONCEPT-i>` are words representing $c$ and its related concepts $c'$. `<ATTRIBUTE>` is the word for an attribute of a object, one of *shape*, *color*, *material* and *size*. |

(c) The test question generation template in CLEVR.

| | |
|---|---|
| **Template** | `<QUEST>` |
| **Example** | Does the small blue thing have the same material as the gray object? |
| **Description** | `<QUEST>` are question generated from the original CLEVR paper Johnson et al. (2017a). |

Table 8: Templates for generating descriptive sentences, supplemental sentences and test questions in CLEVR.

| Model | Example Only | Example+Supp. | $\Delta$ |
|---|---|---|---|
| CNN+LSTM | 41.61 | 42.38 | 0.77 |
| MAC | 62.38 | 62.40 | 0.02 |
| NSCL+LSTM | 57.76 | 61.79 | 4.03 |
| NSCL+GNN | 64.75 | 84.83 | 20.08 |
| FALCON-G | 69.71 | **87.29** | 17.57 |
| FALCON-R | 69.70 | 86.21 | 16.50 |

Table 9: Ablated study of FALCON tasks w/. and w/o. supplemental sentences (+Supp.) on the biased CLEVR dataset. All concept-centric methods use the box embedding space.

On CLEVR, descriptive sentences are generated based on the CLEVR-Ref+ Liu et al. (2019) dataset. Shown in Table 8a, we first generate a referential expression for the target object, and associate it with the novel concept $c$. The referential expression contains only concepts from $\mathcal{C}_{base}$. In total we have generated 7.5k pairs of descriptive sentences and images, 5k of which are for $\mathcal{C}_{base}$, 1k for $\mathcal{C}_{val}$, and another 1.5k for $\mathcal{C}_{test}$. supplemental sentences sentences are generated based on template in Table 8b by combining the concept $c$ and its related concepts $c'$. We generate 225k pairs of testing questions and images based on the templates in Table 8c. 150k of them are associated with concepts in $\mathcal{C}_{train}$, 30k for $\mathcal{C}_{val}$, and another 45k for in $\mathcal{C}_{test}$. Testing question contains only one occurrence of concept $c$ and possibly many occurrences of base concepts. An example of the generated texts can be found in Fig. 5. We generated 402k questions for pre-training.

**Learning from biased data.** We supplement a concrete example to illustrate how our model can learn novel visual concepts from biased data. Shown in Fig. 6, in the first task, the learner is trained on a novel concept *cylinder*. However, all training examples of *cylinders* are also *purple*. Thus, without any supplemental information about whether the novel concept is a color or a shape, the learning task itself is ambiguous. Our model is capable of resolve such kind of ambiguity by interpreting supplemental sentences. We have also included the full experimental result of this setup in Table 9.

**Handling non-prototypical concepts.** On CLEVR, we have studied a specific type of concept-concept relationship: *hypernym*. It is worth noting that there is a difference between the *hypernym* relations in CUB dataset and *hypernym* relations in CLEVR dataset. Specifically, the hypernym of a bird category is also a bird category. However, the hypernym of a concept in CLEVR (e.g., color,

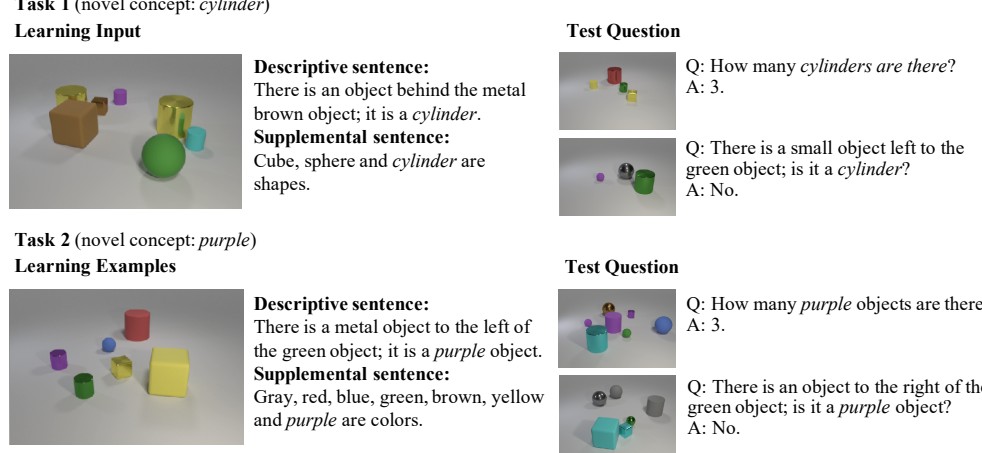

Figure 6: A visualization of fast concept learning from biased data with supplemental sentences on CLEVR. Notice that in both tasks, the visual examples are purple cylinders. The association between the new concept and the visual appearance (shape or color) is ambiguous, without supplemental sentences.

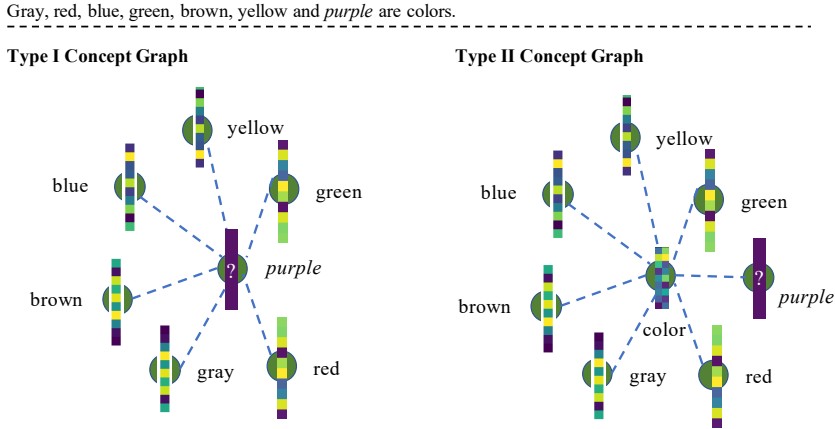

Figure 7: Two implementations of graph $G_{concept}$.

shape, material, etc.) is no longer concepts that can be visually grounded. They are non-prototypical concepts because they can not be associated with a prototypical visual instance.

We have studied two different ways to handle such kind of hypernym relation. They are different in terms of their way of constructing the concept graph $G_{concept}$, namely Type I and Type II. Illustrated in Fig. 7, in Type I graph, the novel concept $c = \mathrm{purple}$ is connected with other concepts that share the same hypernym with $c$. In type II $G_{concept}$, the non-prototypical concept *color* is treated as a node in the concept graph. All concepts that are "colors", including $c$ is connected to this new node *color*. Our experiment shows that, in the fast concept learning test with continual learning setting, our best model (FALCON-G + Box Embedding) achieves a test accuracy of 88.40% using the type I concept graph. With the type II concept graph, it can only reach the accuracy of 71.34%. Thus, throughout the paper, we will be using concept graphs of type I.

## C.3 THE GQA DATASET

In the GQA dataset, descriptive sentences are generated from the referential expressions produced by the templates in Table 10a, containing only concepts from $\mathcal{C}_{base}$. We provided an approximate

| (a) The descriptive sentence template in GQA. |
|---|

| | |
|---|---|
| **Template** | The `<CONCEPT-1>` (`<CONCEPT-2>`) (object) is a/an `<CONCEPT>` (object). |
| **Example** | The small man is a sitting object. |
| **Description** | `<CONCEPT>` is the word for concept $c$. `<CONCEPT-i>` are other concepts. |

| (b) The supplemental sentence template in GQA. |
|---|

| | |
|---|---|
| **Template** | `<CONCEPT-1>`,`<CONCEPT-2>`$\cdots$`<CONCEPT-N>` describes the same property of an object. |
| **Example** | Sitting, standing describes the same property of an object. |
| **Description** | `<CONCEPT-i>` are words representing $c$ and its related concepts $c'$. |

| (c) The test question generation template in GQA. |
|---|

| | |
|---|---|
| **Template** | Is the `<CONCEPT-1>` (`<CONCEPT-1>`) (object) a/an `<CONCEPT>` (object)? |
| **Example** | Is the white woman a sitting object? |
| **Description** | `<CONCEPT>` is the word for concept $c$. `<CONCEPT-i>` are other concepts. |

Table 10: Templates for generating descriptive sentences, supplemental sentences and test questions in GQA.

| Model | QA Accuracy | | |
|---|---|---|---|
| CNN+LSTM | 54.47 | | |
| MAC | 54.99 | | |
| | Box | Hyperplane | Hypercone |
| NSCL+LSTM | 55.41 | 54.19 | 54.81 |
| NSCL+GNN | 55.32 | 50.07 | 54.75 |
| FALCON-G | **55.96** | 54.22 | 54.56 |
| FALCON-R | 55.42 | 55.73 | 55.79 |

| Model | QA Accuracy | | |
|---|---|---|---|
| CNN+LSTM | 48.73 | | |
| MAC | 54.45 | | |
| | Box | Hyperplane | Hypercone |
| NSCL+LSTM | 53.54 | 53.16 | 52.83 |
| NSCL+GNN | 50.15 | 50.74 | 52.03 |
| FALCON-G | 53.89 | 53.24 | 52.22 |
| FALCON-R | **54.60** | 52.86 | 52.37 |

(a) Fast concept learning performance on the GQA dataset. Our model FALCON-G and FALCON-R outperforms all baselines.

(b) Fast concept learning with only visual examples, evaluated on the GQA dataset. In this setting, NSCL+GNN will fallback to a Prototypical Network (Snell et al., 2017).

of 3.9k descriptive sentences, 2.3k of which for $\mathcal{C}_{base}$, 600 for $\mathcal{C}_{val}$ and 1k for $\mathcal{C}_{test}$. Supplemental sentences are generated based on the templates in Table 6b. In supplemental sentences for concept $c \in \mathcal{C}_{base}$, we randomly remove a related concept $c'$ from the sentence with a probability of 0.2. We generate 117k testing questions based on the templates in Table 6c, 13.8k of them paired with concepts in $\mathcal{C}_{train}$, 18k of them paired with concepts in $\mathcal{C}_{val}$ and 30k of them paired with concepts in $\mathcal{C}_{test}$. We generated around 13.4k question-image pairs for pre-training. We use the object-based features produced by a pretrained Faster-RCNN model Ren et al. (2015), which is provided in the dataset.

The main result is listed in Table 11a and 11b. We also compare the model performance on the "few-shot" learning setting, where no supplemental sentences are provided. Out experiments shows that our model (FALCON-G or FALCON-R with box embedding spaces) outperforms other baselines in both settings.

## C.4 THE CONNECTIONS BETWEEN THE BOX EMBEDDING AND OUR FORMULATION

We justify the choice of the box embedding space from two aspects. First, intuitively some supplemental sentences in our setting can be modeled as concept entailment relationships, which can be well modeled by the box embedding space. Second, empirically, the box embedding space, among three embedding space choices studied in the paper, is the best choice for representing entailment.

**Modeling concept entailments (intuition).** In order to learn a new concept $c$, the model receives supplemental sentences relating $c$ with known concepts $c'$. A specific type of relationship is the hypernym relation. The fact that $c$ is a hypernym of $c'$ implies that "all objects that belong to concept $c'$ also belong to concept $c$". In the box embedding space, such relationship can be represented as a

constraint that "the box of $c$ is inside the box $c'$". Thus, the geometric structures of the box embedding space naturally fits the concept entailment relations in our data.

There are two things to be noted. First, there are other non-entailment relations that can not be explicitly written down as a constraint. For example, the "same kind" relationship between concept red and concept blue: they are both colors. Second, instead of explicitly imposing these constraints while inferring the concept embedding for $c$, we use learned models to directly predict the embedding of $c$ - but we hope neural models to leverage such geometric structures.

**Comparison with other box embedding spaces (empirical support).** We have already compared the "box embedding" space with other, commonly-used approaches, such as the "hyperplane space" (i.e., the score is computed by a dot-product), and the "hypercone space" (i.e., the score is computed by cosine-similarity) using downstream task performance measures. The box embedding space outperforms the other two (Table 1 and 2 of the main paper).

Other than downstream task performances, to further validate our intuition, we conduct further analysis on whether the predicted box embeddings of concepts actually satisfy the entailment relations. Specifically, using the CUB dataset, we use trained FALCON-G models based on different embedding spaces to predict the concept embeddings in the validation and the test concept set. For concepts in the training set, we use the learned concept embeddings in the pre-training stage. Next, we use all concept embeddings and the underlying conditional probability measure for each space to compute $\Pr[c'|c]$ for all pairs of concepts $(c, c')$.

We denote $\Pr[c'|c]$ as the prediction score for "$c$ entails $c'$" (i.e., the hypernym relation: $c'$ is a hypernym of $c$). In particular, we use a family of classifiers $entails_t(c, c') = \mathbf{1}[\Pr[c'|c] > t]$, where $\mathbf{1}[\cdot]$ is the indicator function and $t$ a threshold parameter. The conditional probability $\Pr[c'|c]$ is defined as the following. Denote $e_c$ and $e_{c'}$ as the concept embeddings for $c$ and $c'$.

- For the box embedding space, $\Pr[c'|c] = \Pr[e_{c'} \cap e_c] / \Pr[e_c]$, where $\Pr[e_{c'} \cap e_c]$ and $\Pr[e_c]$ have been defined in Section 3.1.

- For the hyperplane embedding space, $\Pr[c'|c] = \sigma(e_c^T e_{c'} / \tau - \gamma)$, where $\sigma$ is the sigmoid function, $\tau = 0.125$, and $\gamma = 2d$ are the same set of scalar hyperparameters used for classifies objects. $d$ is the embedding dimension.

- For the hypercone embedding space, $\Pr[c'|c] = \sigma((\langle e_{c'}, e_c \rangle - \gamma)/\tau)$, where $\langle \cdot, \cdot \rangle$ is the cosine similarity function, and $\tau = 0.1, \gamma = 0.2$ are the same set of scalar hyperparameters used for classifying objects.

By varying the threshold $t$, we compute the AUC-ROC score for FALCON-G based on three different embedding spaces, shown in Table 12. This results suggest that the predicted concept embeddings from FALCON-G (Box) indeed better capture the hypernym relations between concepts. Note that the choice of specific hyperparameters: $\tau$, $\gamma$, etc., will not affect the AUC-ROC score. We also want to highlight that the hyperplane and hypercone embedding spaces do not naturally support a conditional probability measure for two concepts, and thus are not optimized for capturing concept entailments, yielding a lower AUC-ROC score.

| Methods | FALCON (Box) | FALCON (Hyperplane) | FALCON (Hypercone) |
|---|---|---|---|
| AUC-ROC Score | 0.855 | 0.638 | 0.585 |

Table 12: AUC-ROC score of classifying entailment relationship based on concept embeddings from different embedding spaces.

# D ABLATION DETAILS

This section discusses two ablation studies on how the number of base concepts in pretraining and the effect of the number of related concepts in supplemental sentences.

| # of base concepts | FALCON-G | NSCL+GNN | MAC |
|:---:|:---:|:---:|:---:|
| 130 | 76.32 | 75.21 | 65.16 |
| 211 | 81.33 | 78.50 | 73.88 |

Table 13: Fast concept learning (with supplemental sentence) performance under different number of base concepts.

| % of all related concepts | FALCON-G | NSCL+GNN | MAC |
|:---:|:---:|:---:|:---:|
| 0 | 76.37 | 73.38 | 73.55 |
| 25 | 80.20 | 77.16 | 73.94 |
| 50 | 80.78 | 76.93 | 74.13 |
| 75 | 81.20 | 77.77 | 74.23 |
| 100 | 81.33 | 78.50 | 73.88 |

Table 14: Fast concept learning performance under different percentage of related concepts in the supplemental sentence.

### D.1 THE EFFECT OF THE NUMBER OF BASE CONCEPTS

In this section, we will conduct an ablation study on how the number of base concepts can contribute to the model performance. We design a new split of the CUB dataset derived from 50 training species (130 base concepts), 50 validation species (81 concepts), and 100 test species (155 concepts). In our original experimental setup, we have used 100 training concepts (211 base concepts).

We perform the same fast concept learning experiments on this new split, using our model FALCON-G, a concept-centric baseline NSCL+GNN, and an end-to-end baseline MAC. All concept-centric models use box embeddings spaces.

Our results are summarized in Table 13. Since we leverage the relationship between the novel concept and known concepts during the learning of the novel concept, all methods have an accuracy drop when there are fewer base concepts. This demonstrates that transfering knowledge from concepts already learned is helpful. Moreover, our model FALCON-G still has the highest accuracy when the number of base concepts is reduced.

### D.2 THE EFFECT OF THE NUMBER OF RELATED CONCEPTS IN SUPPLEMENTAL SENTENCES

In this section, we design an ablation study on how the number of supplemental sentences would affect model performance. Since all information captured in the supplemental sentences are the names of related concepts and their relations with the novel concept, we provide an ablation on the effect of the number of related concepts in supplemental sentences.

In the following experiments, we evaluate several models on the fast concept learning tasks on the CUB dataset. For each model, we use supplemental sentences containing 0% (no supp. sentence), 25%, 50%, 75%, and 100% (the setting described in the main paper) of all related concepts. Again, we compare our model FALCON-G, a concept-centric baseline NSCL+GNN, and an end-to-end baseline MAC. All concept-centric models use box embeddings spaces.

The results are summarized in Table 14 and Fig. 8. We have the following observations.

1. More related concepts included in the supplemental sentence generally lead to higher accuracy, across all models.

2. In both ceoncept-centric models (FALCON-G and NSCL+GNN), the most significant improvement in test accuracy occurs between 0% and 25%. This suggests that these models can benefit from even just an incomplete set of related concept information.

3. Our model, FALCON-G, performs consistently the best across all percentages of related concepts.

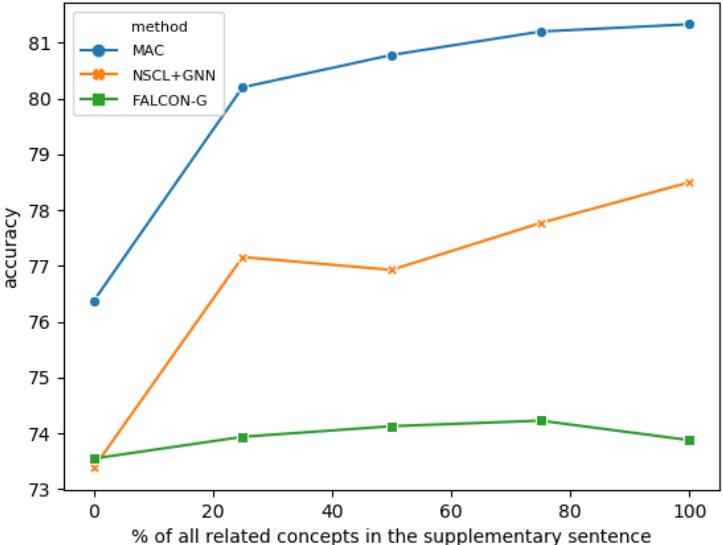

Figure 8: Fast concept learning performance under different percentage of related concepts in the supplemental sentence.

# E  FAILURE MODE DETAILS

This section discusses how failure may occur in the detection module, the semantic parser, and the concept learning module, which may or may not result in a wrong answer.

## E.1  FAILURE MODE IN DETECTION MODULE

There are a few circumstances where the detection module from the feature extractor produces an error. Here we list two cases where the detection module may fail, illustrated in Fig. 15. In one scenario, the model may fail to detect one object from the scene, as in Fig. 15a. Such errors usually occur among small and partially occluded objects. In the latter scenario, the model may recognize one object mask that does not correspond to any object, creating a false positive, as in Fig. 15b. However, the example object may be referenced correctly, as the error does not affect the evaluation of the program in the neural symbolic execution phase. In all scenarios, the detection error leads to an incorrect object or a semantically ambiguous object being referenced during program evaluation. Such error may propagate to the meta-learning stage, change the novel concept's embedding, and eventually result in a correct or wrong answer.

## E.2  FAILURE MODE IN SEMANTIC PARSER

There are a few circumstances where the semantic parser produces an error. Here we list three cases where the semantic parser may fail, illustrated in Table 16. First, the model may produce a synthetically wrong program. For example, there are invalid concept arguments in the output program. Following Johnson et al. (2017b), the parser applies heuristics to the output, including removing the invalid subprograms where concept arguments are illegal, and produces a feasible program, as in Table 16a. In other circumstances, the model produces a synthetically correct but semantically wrong program. We will evaluate the program as it is produced by the Seq2seq. Such evaluation will either lead to the right answer or lead to the wrong answer. It should be no surprise that a semantically incorrect program may lead to a wrong answer since the concept arguments are wrong, as in Table 16b. A semantically wrong program may also lead to the correct answer if the question has redundancy in concept quantifiers that refers to the correct object, as in Table 16c. Currently, there is no mechanism implemented trying to recover from the semantic errors. In the future, we

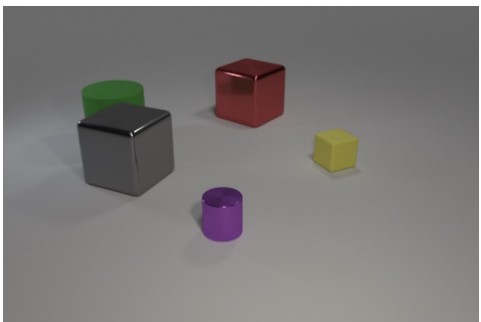 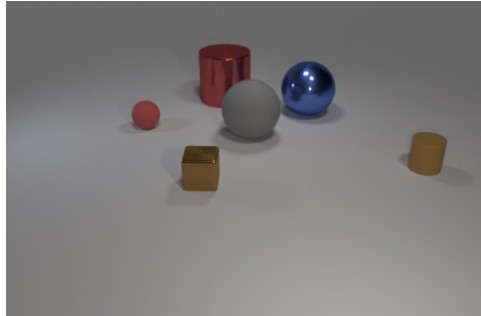

**Descriptive sentence**: The shiny block that are behind small rubber thing is a red object.
**Expected example object**: The metallic red cube.
**Predicted example object**: The metallic grey cube.
**Explanation**: The relational concept *behind* is not accurate enough; it is so extensive that even the grey cube is behind the yellow cube.

(a) Error case when the imperfect visual grounding leads to a wrong object example in the training image. This is part of the task that learns the novel concept *red*.

**Question**: How many red cylinders are to the right of the blue sphere?
**Expected answer**: 1. (The red cylinder)
**Predicted answer**: 0.
**Explanation**: The meta-learned concept *cylinder* is not accurate enough; it is too exclusive that even the red cylinder is not considered a *cylinder*.

(b) Error case when the imperfect visual grounding leads to a wrong answer in downstream question-answering. This is part of the task that learns the novel concept *cylinder*.

Figure 9: A visualization of two failure cases of the concept learning module.

hope to incorporate methods such as REINFORCE (as in Yi et al. (2018) and Mao et al. (2019)) to leverage visual information to refine the semantic parser.

### E.3 FAILURE MODE IN CONCEPT LEARNING MODULE

Besides errors in the semantic parser and the detection module, the concept learning module may also lead to potential errors due to imperfect visual groundings. Here we list two cases where the concept learning module may fail, illustrated in Fig. 9. In the first scenario, the model encounters an error when filtering for a relational concept. The concept in the program is too extensive and filters out more candidates than desired. Such error results in a wrong object example, as in Fig. 9a. In another scenario, the model encounters an error when filtering for a novel concept recently meta-learned. The concept in the program is too exclusive and filters out fewer candidates than desired. Such error results in a wrong answer in the final result, as in Fig. 9b. In general, errors may take place in the concept learner when filtering for a base concept, a relational concept, or the novel concept just meta-learned when the visual grounding is imperfect. Such errors may result in different filter results with more or fewer objects being included. Further operation on the wrong filter output may eventually end up with a correct or wrong answer.

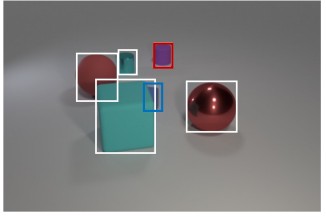

**Descriptive sentence**:The blue matte thing to the right of the big cyan object is a cube.
**Expected example object**: The blue cube around the upper-right corner of the cyan cube.
**Predicted example object**: The purple metal cylinder. ✗
**Detection error**: The blue cube is not detected by the detection module.

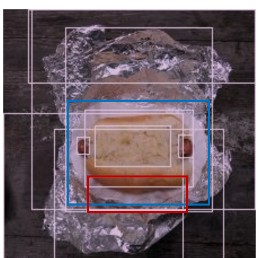

**Descriptive sentence**: The white small object is a plate.
**Expected example object**: The white plate.
**Predicted example object**: The lower side of the plate. ✗
**Detection error**: The entirety of the white plate is not detected as one object

(a) Error case when the detection module misses an object. The blue boxes indicates the expected object that the detection module misses; the red boxes indicates the wrongly predicted objects.

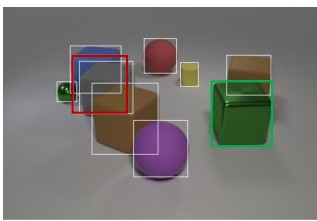

**Descriptive sentence**:The metallic thing that is right of the tiny sphere is a cube.
**Expected example object**: The green cube.
**Predicted example object**: The green cube. ✓
**Detection error**: The union of the blue cube and the gray cube is detected as one object by the detection module.

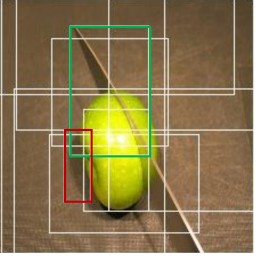

**Descriptive sentence**:The large long object is a silver object.
**Expected example object**: The long knife.
**Predicted example object**: The long knife. ✓
**Detection error**: Shadows of the knife and the apple are detected as objects by the detection module.

(b) Error case when the detection module recognize a false positive as an object. The green boxes indicates the expected and predicted object; the red boxes indicates false positives.

Table 15: A visualization of three failure cases of the detection module. The first two descriptive sentences are evaluated to find the wrong example object, and the last two are evaluated to find the right example object.

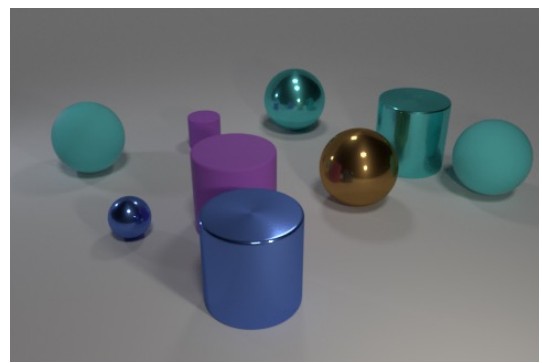

| Text | Are there an equal number of small balls in front of the small ball and big cyan things? |
|---|---|
| **Expected Program** | ```CountEqual(Filter(Filter(Relate(```
```Filter(Filter(Scene(), small), ball),```
```front), small), ball),```
```Filter(Filter(Scene(), big), cyan))``` |
| **Predicted Program** | ```CountEqual(Filter(Filter(Relate(```
```Filter(Filter(Scene(), small), ball),```
```front), small), Filter),```
```Filter(Filter(Scene(), big), cyan))``` |
| **Fixed Program** | ```CountEqual(Filter(Relate(```
```Filter(Filter(Scene(), small), ball),```
```front), small),```
```Filter(Filter(Scene(), big), cyan))``` |

(a) A syntactically infeasible program. The program is fixed with heuristics.

| Text | Is there a metal cyan object to the left of the big brown object? |
|---|---|
| **Expected Program** | ```Exists(Filter(Filter(Relate(```
```Filter(Filter(Scene(), big), brown),```
```left), metal), cyan))``` |
| **Predicted Program** | ```Exists(Filter(Filter(Relate(```
```Filter(Filter(Scene(), big), purple),```
```left), metal), cyan))``` |
| **Expected answer**  Yes.  **Predicted answer**  No. | |

(b) A syntactically feasible but semantically wrong program, which yields a wrong answer.

| Text | Is there a big cyan object to the right of the big brown object? |
|---|---|
| **Expected Program** | ```Exists(Filter(Filter(Relate(```
```Filter(Filter(Scene(), big), brown),```
```right), big), cyan))``` |
| **Predicted Program** | ```Exists(Filter(Filter(Relate(```
```Filter(Filter(Scene(), big), purple),```
```right), big), cyan))``` |
| **Expected answer**  Yes.  **Predicted answer**  Yes. | |

(c) A syntactically feasible but semantically wrong program, but the answer is correct.

Table 16: A visualization of three failure cases of the semantic parser. All examples are from the question answering part of tasks on learning the novel concept *cyan* based on the image above.

