# OpenReview forum: "FALCON: Fast Visual Concept Learning by Integrating Images, Linguistic descriptions, and Conceptual Relations"
_ICLR.cc/2022/Conference — ICLR 2022 Poster_

### Official Review · Reviewer_98FU · 2021-10-21

**Correctness:** 3
**Technical Novelty And Significance:** 3
**Empirical Novelty And Significance:** 2
**Recommendation:** 6
**Confidence:** 3

**Main Review:**

The paper is very clear to read. It is a relatively novel task to learn the new visual concepts from a few images and sentences. In the experiments, the authors validate their methods in both real-world and synthetic datasets.

The model that the author provides contains a few separate components, such as some semantic parsers, that cannot be jointly optimized. The proposed method cannot recover from the errors made by those components.

It would be great to explain in a bit more details about the failure patterns of the proposed method.


**Summary Of The Paper:**

The paper provides a method for learning new concepts from a few examples with both images and natural languages. For the evaluation, the proposed model is able to answer questions about some images related to the visual concepts.


**Summary Of The Review:**

I find the paper interesting to read, and believe there are enough novelties in the proposed paper. I recommend a rating of “weakly accept”.

---

> ### Author Response · Authors · 2021-11-20
> **Response to 98FU**
>
> Thank you for your insightful review and helpful feedback!  In the following, we aim to address your specific concerns about our failure cases. Please refer to the general response and the appendix section of the updated paper for failure case analysis in these two model components.
>
> **Q1:** Failure cases of our semantic parser.
>
> **A1:** We agree that the semantic parser is not optimized jointly with the rest of the model under the current framework, thus leading to a potential risk of unrecoverable failure. In the future, we hope to incorporate methods such as REINFORCE (as in Yi et al. [1] and Mao et al. [2]) for joint learning. We have uploaded a revised version of the paper to include more failure case analysis of the semantic parser module and how it can affect the overall performance. Please refer to the general response for more details.
>
> **Q2:** Failure patterns.
>
> **A2:** Thank you for the suggestion. We have summarized several possible causes of failure in detail. These failures can be attributed to the errors incurred in the object detection phase, in the semantic parsing phase, or the program execution phase. Please refer to the general response section and the updated appendix of the paper.
>
> Again, thank you for your comments, and we hope that our response addresses your concerns! If you have any additional questions, please feel free to let us know during the rebuttal window.
>
> **References**
>
> [1] Kexin Yi, Jiajun Wu, Chuang Gan, Antonio Torralba, Pushmeet Kohli, and Joshua B. Tenenbaum. Neural-Symbolic VQA: Disentangling Reasoning from Vision and Language Understanding. In NeurIPS, 2018.
>
> [2] Jiayuan Mao, Chuang Gan, Pushmeet Kohli, Joshua B. Tenenbaum, and Jiajun Wu. The Neuro-Symbolic Concept Learner: Interpreting Scenes, Words, and Sentences From Natural Supervision. In ICLR, 2019.

---

> > ### Comment · Reviewer_98FU · 2021-11-30
> > **Response to the rebuttal**
> >
> > Thank you for the added examples and future plans! I am generally satisfied with the answer and will keep my rating to be accept.

---

### Official Review · Reviewer_6cyF · 2021-10-31

**Correctness:** 4
**Technical Novelty And Significance:** 3
**Empirical Novelty And Significance:** 4
**Recommendation:** 8
**Confidence:** 4

**Details Of Ethics Concerns:**

The paper shares common concerns on machine learning systems (fairness, biases, etc), but there seems no extra specific concern in this work.

**Main Review:**

## Strength

The paper describes a novel technical approach to fast concept learning from a few examples. The proposed framework integrates the neuro-symbolic program executor, box embeddings, and graph neural networks to achieve the competitive performance in the considered benchmarks.

The main technical idea seems to lie in the use of the box embedding in the concept-centric (neuro-symbolic) approach. Section 2 concisely summarizes the context of this work.

The evaluation protocol looks comprehensive and convincing, including both detailed ablation and SoTA comparisons in multiple datasets.

## Weakness

I do not have a major concern on this paper. Below are a few minor questions.

The paper uses a pretrained Mask R-CNN for extracting objects in a given image. I wonder what is the effect of detection errors and the influence of on what dataset these detectors are trained. Presumably the neuro-symbolic reasoning module should be able to handle errors, but it seems possible to completely miss a correct object in the scene.

Another similar question is the error in language parsing. I assume the generator in this work does not incur an error in semantic parsing thanks to the use of templates, but was there any error originating from parsing / program execution?

**Summary Of The Paper:**

The paper describes an approach to learn visual concepts with a few examples. The paper considers a problem consisting of paired images and sentences, optional concept descriptions, and the target question answering task for a new concept. The proposed approach utilizes the object detector (He 2017) to map visual objects or relationship into the box embedding space (Vilnis 2018), and also utilizes a neuro-symbolic program to identify the referring object as well as relationship among concepts. The new concept representation is inferred after the two graph neural networks, and the networks are trained to solve for the downstream task, which in this paper is question answering. The proposed approach is evaluated in three benchmarks (CUB, CLEVR, GQA) and shown to outperform the recent baselines (Hudson &Manning 2018, Mao 2019).

**Summary Of The Review:**

The paper proposes a novel framework for fast visual concept learning that builds on a neuro-symbolic program executor, box embeddings, and graph neural networks. Even if the individual components might be from the existing work, the overall approach presents a novel technical approach. The evaluation based on downstream question answering shows the competitive performance of the proposed model over baselines in multiple benchmarks. Given the novelty and the effectiveness, I believe the paper makes a solid contribution.

---

> ### Author Response · Authors · 2021-11-20
> **Response to 6CYF**
>
> In the following, we will address your concerns on the failure cases of the detection module and the semantic parser. Please refer to the general response and the appendix section of the updated paper for failure case analysis in these two model components.
>
> **Q1:** Failure cases of our pre-trained detection module.
>
> **A1:** Thank you for the question. We agree that using a pre-trained detection module may result in unrecoverable errors that can propagate to the later stage of program execution. For more details of the failure case analysis of the detection module, please refer to the general response section and the failure case analysis section of the updated appendix.
>
> **Q2:** Failure cases of our semantic parser.
>
> **A2:** We agree that the semantic parser is not optimized jointly with the rest of the model under the current framework, thus leading to a potential risk of unrecoverable failure. In the future, we hope to incorporate methods such as REINFORCE (as in Yi et al. [1] and Mao et al. [2]) for joint learning. We have uploaded a revised version of the paper to include more failure case analysis of the semantic parser module and how it can affect the overall performance. Please refer to the general response for more details.
>
> Again, thank you for your comments, and we hope that our response addresses your concerns! If you have any additional questions, please feel free to let us know during the rebuttal window.
>
> **References**
> [1] Kexin Yi, Jiajun Wu, Chuang Gan, Antonio Torralba, Pushmeet Kohli, and Joshua B. Tenenbaum. Neural-Symbolic VQA: Disentangling Reasoning from Vision and Language Understanding. In NeurIPS, 2018.
>
> [2] Jiayuan Mao, Chuang Gan, Pushmeet Kohli, Joshua B. Tenenbaum, and Jiajun Wu. The Neuro-Symbolic Concept Learner: Interpreting Scenes, Words, and Sentences From Natural Supervision. In ICLR, 2019.

---

> > ### Comment · Reviewer_6cyF · 2021-11-30
> > **Final rating**
> >
> > Thanks for the additional ablation and error analysis. I believe the paper has a solid contribution as commented in the initial review. I would recommend an accept in the final rating.

---

### Official Review · Reviewer_W3YC · 2021-11-02

**Correctness:** 4
**Technical Novelty And Significance:** 3
**Empirical Novelty And Significance:** 3
**Recommendation:** 8
**Confidence:** 4

**Main Review:**

The paper's idea is very interesting and novel. I really enjoy reading this paper. I have some questions for the specific details and the datasets.

For the dataset (sect. 4.1), if I understand the approach correctly, the approach first needs a paired image and sentence to learn the base concepts. In this setting, the sentence should be a descriptive sentence that describes the object/concept in the image. If my understanding is correct, the CUB dataset setting is very natural. However, I wonder how would CLEVR and GQA dataset follow this setting? IIUC, the text in CLEVR and GQA are all questions, which are not descriptive sentences.

For the approach (sect. 3.3), IIUC, the proposed approach needs to extract concept relations from a supplemental sentence. First, the sentence is in the natural language form. How to do the semantic parsing of the sentence? The parsing might contain error. How to resolve that? As the sentence is in the natural language form, the name of the concept might in different form. How to resolve that? Second, it is pretty rare to see a paper that makes neural symbolic machine works for the real language. The neural symbolic machine works very well for the template language. Then I wonder how to design the operators in the neural symbolic machine?

For the setting, I wonder what is the setting for CUB? IIUC, CUB is a dataset for bird classification. From Fig. 4, it seems this paper treat the CUB as a retrieval task? The images of CUB dataset only contains one single bird. Therefore, I think the object centric approach might works pretty well. (Optional, this might be a stretch) I wonder would this approach works for some other datasets like Flickr30k or COCO?

**Summary Of The Paper:**

This paper tackles concept learning problem. This paper designed a neural symbolic machine based approach that is able to inference the concept embedding from a given image and sentence pair. It can also mediate the concept embedding using additional text explanation. To verify this approach, this paper follows a meta-learning setting. Specifically, the concepts are split into training, validation, and testing. The model first learn the base concepts from the training set and then quickly infer the concept embedding for the testing split. The paper achieved a superior performance over three datasets against baselines.

**Summary Of The Review:**

I think this paper is a very interesting paper for concept learning from image and natural language. The motivation of this paper is very natural and the architecture is chosen appropriately. As long as the author could address my questions for the setting, datasets, and details of the approach, I would recommend accept.

---

> ### Author Response · Authors · 2021-11-20
> **Response to W3YC**
>
> Thank you for your insightful review and helpful feedback! In the following, we address your question concerning our dataset and other specific details.
>
> **Q1:** Generation of descriptive sentences in CLEVR and GQA.
>
> **A1:** Yes, we have generated descriptive sentences for the CLEVR and the GQA datasets. In CLEVR, the descriptive sentences are based on the templates used by CLEVR-Ref+ (Liu et al., [1] which is a referring expression dataset). Just as how CLEVR generates questions, our descriptive sentence generator takes the groundtruth concept annotations and produces sentences such as "There is a small object to the right of the big yellow cube; it is a cyan object." Similarly, we generate descriptive sentences for GQA. For more details, please refer to the dataset generation section of the appendix in our paper.
>
> **Q2:** The process of semantic parsing and error handling.
>
> **A2:**: We have added new details in the appendix about our semantic parsing. Here we briefly explain.
>
> Our semantic parser follows the general framework of Seq2seq (Sutskever et al. 2014 [2]) and our implementation follows NS-VQA (Yi et al. [3]), which has been developed for VQA tasks.
> We first tokenize the input sentence into a sequence of word ids. Then, we feed the sequence into a Seq2Seq model and generate a sequence of $(op_i, concept_i)$, where $i=1,2,3,...,L_{out}$ where $L_{out}$ is the output sequence length. $op_i$'s are operations defined in the DSL, and they form a "flattened" program based on which we will recover a hierarchical program. Some operations such as $filter$ has a concept input, in which case we use the corresponding $concept_i$. The semantic parser is trained on 10% of all generated data.
>
> We recognize that there are a few circumstances where the semantic parser produces an error. Since our semantic parser is pre-trained, there are parsing errors that can not be recovered during the program execution. For more details, please refer to the general clarification section and the failure analysis section of the updated paper.
>
> **Q3:** The name of the concept might in different forms.
>
> **A3:** In the concept-centric learning models (FALCON and NSCL), different word forms of the same underlying concept (e.g., cube and box in CLEVR) are treated as distinct concepts and learned separately. Thus, when a novel concept appear in a sentence, we will directly use the word form to initialize a novel concept emebdding.
>
> **Q4:** Operations in Real-World Languages.
>
> **A4:** We agree with the reviewer that real-world language contains more complex structures and also richer semantics. First, we think the "filter" (find all objects that have a property) and "relate" (find all pairs of objects that have a relation) already have a significant coverage of word meanings in real-world language, especially in captions (see Anderson et al. [4]). Second, there are certain operations missing in the current CLEVR/CUB/GQA datasets. One examples is quantifiers (some of .../all of ...). Third, in real-world language, people may communicate about concepts beyond object properties and relations, such as the intention of human in images. This may require additional operations too.
>
> **Q5:** Figure 4 and the setting on CUB.
>
> **A5:** Our setting on CUB is also a visual question answering setting. In figure 4, we are showing the answer to the question (Is there a sturnella?) on six different images (and the confidence score of the model).

---

> > ### Author Response · Authors · 2021-11-20
> > **Response to W3YC - continued**
> >
> >
> > **Q6:** Extension to other dataset.
> >
> > **A6:** Thank you for the suggestion. Extending to other datasets such as Flickr30k and COCO is an interesting potential direction for our model. We outline two challanges.
> >
> > First, the evaluation. Both datasets do not provide annotations for the concepts appeared in the captions (MS-COCO [5] has bouding boxes for certain objects but is considerably smaller than the concepts mentioned in captions). Thus, it will be hard to separate the image set into training and test splits. This is a primary reason for us to choose GQA (which contain concept annotations from visual genome) as one of our real-world image benchmarks.
> >
> > Second, another challenge is to build models that generalizes to real-world languages. See also our A4 for a discussion.
> >
> > Again, thank you for your comments, and we hope that our respnse addresses your concerns! If you have any additional questions, please feel free to let us know during the rebuttal window.
> >
> > **References**
> >
> > [1] Runtao Liu, Chenxi Liu, Yutong Bai, and Alan L. Yuille. CLEVR-Ref+: Diagnosing Visual Reasoning With Referring Expressions. In CVPR, 2019.
> >
> > [2] Ilya Sutskever, Oriol Vinyals, and Quoc V. Le.  Sequence to Sequence Learning with Neural Networks.  In NeurIPS, 2014.
> >
> > [3] Kexin Yi, Jiajun Wu, Chuang Gan, Antonio Torralba, Pushmeet Kohli, and Joshua B. Tenenbaum. Neural-Symbolic VQA: Disentangling Reasoning from Vision and Language Understanding. In NeurIPS, 2018.
> >
> > [4] Peter Anderson, Basura Fernando, Mark Johnson, and Stephen Gould. SPICE: Semantic Propositional Image Caption Evaluation. In ECCV, 2016.
> >
> > [5] Tsung-Yi Lin, Michael Maire, Serge Belongie, Lubomir Bourdev, Ross Girshick, James Hays, Pietro Perona, Deva Ramanan, C. Lawrence Zitnick and Piotr Dollár. Microsoft COCO: Common Objects in Context. In ECCV, 2014.

---

> > > ### Comment · Reviewer_W3YC · 2021-12-02
> > > **Final Rating**
> > >
> > > Thanks for the thorough and insightful response. The author resolved my questions. Generalizing to real-world image and language is challenging. I think this paper provide an interesting approach towards that. I would recommend 8.

---

### Official Review · Reviewer_2P5Z · 2021-11-04

**Correctness:** 3
**Technical Novelty And Significance:** 2
**Empirical Novelty And Significance:** 2
**Recommendation:** 5
**Confidence:** 3

**Main Review:**

Overall, I felt like this paper has potential but it just is not there yet. The writing and paper structure needs work. It took a really long to get to the true contribution of the paper, which if appears to be a new training and evaluation setup to evaluate how well a model can answer questions about novel concepts. The problem setup involves a meta-learning step where the model is presented with the novel concept in the form of a sentence and an image along with optional supplementary sentences. Aside from the problem formulation, the paper proposes a concept learning module on top of the Han et al. 2019’s Visual concept meta-concept learning model. The concept learning module is introduces as either a graph neural network or an RNN architecture that learns to relate the concepts extracted from the images, sentences and supplementary sentences.

All older concepts are learned during a pretraining stage (the one introduced in Han et al.). A new addition is the box embedding space that improves performance of the embedding space.


The problem I have with the paper is that it introduces (1) a new problem formulation, (2) a new meta-learning model, and (3) uses a geometric embedding space. But none of these are really justified.

I would have expected to see a paper that answers: (1) Why is this an ecologically valid problem formulation? How does it relate to existing problem formulations? Why do existing baselines fail? (2) why a meta-learning approach is the best formulation to tackle the problem? And (3): Why the box embedding space? There should be more justification for by the box embedding space is the right one to use for the given data / task. The paper has the following sentence: “Here, we focus on the box embedding space (Vilnis et al., 2018) since it naturally models the entailment relationships between concepts.” but doesn’t dive into what these entailments are.

Aside from this, I didn’t find any ablation that studies how the model performs as the number of supplementary sentences changes. How does it scale as the number of base concepts changes or the number of object categories? Is there a cold start problem of requiring a certain number of base concepts during pre-training?



Minor:
- There are a few typos and incomplete sentences. Ex, “Our goal is to use meta-learning a system that can learn new concept quickly.” and “4) use the learned concept flexibly in different concepts”.

There are too many terms that all sound the same in the paper: “relational concepts”, “relational representations of objects”, “object-based concepts”, etc. It took me a while to wrap my head around which ones mean what. I suggest using consistent terminology throughout the paper when referring to a specific concept.


**Summary Of The Paper:**

This paper presents a unified meta-learning neuro-symbolic framework for fast visual concept learning from diverse data streams. It introduces a new embedding prediction module to integrate visual examples and relations to infer novel concept embeddings. It’s meta-learning continuous learning approach also uses supplementary sentences to relate concepts to one another. They show improvements for the end task of question answering with novel concepts by first meta-learning those concepts during a meta-testing phase.


**Summary Of The Review:**

Overall, I think the contribution would have been stronger with a proper grounding of why the problem is ecologically valid, why the solution proposed is an appropriate one, and why specific design choices (box embedding) were made. There are also missing ablations and rigorous evaluation of the model.

---

> ### Author Response · Authors · 2021-11-20
> **Response to 2P5Z**
>
> Thank you for your insightful review and helpful feedback! We agree that the proposed clarifications and ablation studies should be added to the paper for better clarity and completeness. We are happy that we are able to address all of the reviewer's requested ablations. In the following, we aim to address your specific concerns about our formulation and baselines, our choice of approaches that use meta-learning and box embeddings, and various factors that could impact model performance.
>
> **Q1:** Why is our formulation ecologically valid?
>
> **A1:** We believe that it is important to build machines that can learn concepts that are associated with the physical world in an incremental manner and flexibly use them. Language provides a natural interface for doing so: human can teach machines new concepts using language and referring to objects in a scene, and after learning, machines can respond to human queries.
>
> In this paper, we focus on visually grounded language. The machine first looks at an image and hears the utterance “the shape of the object on the table is called dax.” The machine needs to 1) interpret the referring expression “the object on the table” based on the words it knows (object, on, table) and 2) learn a representation for the novel word (dax) by leveraging other prior knowledge it has learnt (other shapes). After that, we 3) test the machine on new scenes and new questions, e.g., is there a dax on the chair? A framework that can solve this challenge will allow us to build machines that can better learn from human and communicate with human. We have better clarified the motivation in the revision.
>
> **Q2:** How is our formulation related with existing ones?
>
> **A2:** We have updated the related work section to make these more clear. To the best our knowledge, we are unaware of any existing work that studies same “fast visual concept learning” setting as we do, but, of course, our problem formulation is inspired by many existing works, such as metaconcept learning (Han et al. [1]), meta-learning (Kampffmeyer et al [2]), and few-shot learning (Snell et al. [3]). In particular, for example, Han et al have shown how jointly learning concepts and metaconcepts can help each other. Our work is an novel approach towards making use of the metaconcepts in a meta-learning setting aiming at boost the learning of novel concepts based on known ones.
>
> **Q3:** Why do baselines fail?
>
> **A3:** We have updated the experiments section to make the following analysis more clear. Below are our key findings, which further justifies our model design.
> * Comparing end-to-end neural baselines (CNN+LSTM, MAC) v.s. explicit concept representations and neuro-symbolic reasoning (NSCL+$*$, FALCON), we show that using explicit concept representation is helpful for complex descriptions and questions (CLEVR, Table 3).
> * Comparing modules for predicting concept embeddings from supplemental sentences $D_c$: LSTMs over texts $D_c$ (NSCL+LSTM) v.s. GNNs over concept graphs $G_{concept}$ (NSCL+GNN, FALCON-G), we see that, explicitly representing supplemental sentences as graphs and using GNNs to encode the information yields the larger improvement
> * Comparing modules for predicting concept embeddings from visual examples $o_i^{(c)}$: heuristically taking the average (NSCL+GNN) v.s. GNNs on example graphs (FALCON-G), we see that using example graphs further improves the performance (Table 2 and Table 3 Example Only, also see our general response for the results on two new ablation studies).
>
> **Q4:** Why is metalearning a good formulation?
>
> **A4:** Ultimately, we want to build systems that can incrementally learn new concepts from grounded language.
> There are two options to approach this problem. One is manually specifying rules to compute the representation for the new concept (in NSCL+GNN, we use the average embedding of the instances of the new concept). The other one is by meta-learning (learning a model to predicts the embedding for the new concept. We show that meta-learning-based approaches are doing better.
> In evaluation, our evaluation follows many meta-learning approaches (e.g., Snell et al. [3]). Specifically, we use different splits of the dataset, feed part of the concepts as “training concepts,” and test model’s fast learning performance on “novel concepts.”

---

> > ### Author Response · Authors · 2021-11-20
> > **Response to 2P5Z, continued**
> >
> > **Q5:** The box embedding space and “entailment.”
> >
> > **A5:** We have updated the paper description to include examples. Box embedding space represents each concept as a high-dimensional cube in the latent space. Following Vilnis et al. [4], it supports a probabilistic interpretation of “entailment” P(A->B), that is, if an objet is of type A (e.g., Laysan Albatross), it must also be of type B (e.g., Albatross). Intuitively, this corresponds to the case where the box A is contained in box B. We hope it can better represents the relationships between different categories, such as brids.
> >
> > We have also compared the “box embedding” space with other existing approaches, such as the “hyperplane space” (i.e., the score is computed by a dot-product), and the “hypercone space” (i.e., the score is computed by cosine-similarity). The box embedding space outperforms the other two (Table 1 and 2).
> >
> > **Q6:** The effect of the number of base concepts and the number of related concepts in supplemental sentences.
> >
> > **A6:** Thank you for the great suggestion. We have added two ablation studies in the paper appendix. Please refer to our general response section for details.
> >
> > Again, thank you for your comments, and we hope that our response has addressed all of your concerns  and turns your assessment to the positive side. Please don’t hesitate to let us know if you want any further clarifications.
> >
> > **References**
> >
> > [1] Chi Han, Jiayuan Mao, Chuang Gan, Joshua B. Tenenbaum, and Jiajun Wu.  Visual Concept Metaconcept Learning. In NeurIPS, 2019.
> >
> > [2] Jake Snell, Kevin Swersky, and Richard S. Zemel. Prototypical Networks for Few-shot Learning. In NeurIPS, 2017.
> >
> > [3] Michael Kampffmeyer, Yinbo Chen, Xiaodan Liang, Hao Wang, Yujia Zhang, and Eric P Xing. Rethinking knowledge graph propagation for zero-shot learning. In CVPR, 2019.
> >
> > [4] Luke Vilnis, Xiang Li, Shikhar Murty, and Andrew McCallum. Probabilistic Embedding of Knowledge Graphs with Box Lattice Measures. In ACL, 2018.

---

### Author Response · Authors · 2021-11-20
**General Response**

We thank the reviewers for their careful reading, and detailed and considerate feedback. We are glad that reviewers mostly agree that FALCON is a novel and well-motivated framework for fast concept learning that combines known concepts and very few additional data ("very interesting and novel", "very natural" (W3YC), "solid contribution" (6CYF), "relatively novel task" (98FU)). We are further glad that the reviewers mostly agree with our solution to this framework that combines neural symbolic reasoning with meta-learning to learn the novel concept ("chosen appropriately" (W3YC), "novel technical approach" (6CYF),"enough novelties" (98FU)), and that the paper is mostly well written ("enjoy reading the paper" (W3YC), "comprehensive and convincing" (6CYF), "very clear to read"(98FU)).

The reviewers also agree that further ablations will strengthen the paper, highlighting its strengths, clarifying limitations, and outlining essential directions for future work. We agree, and we are happy to include the suggested experiments---see below and also our updated manuscript. In our general response, we would like to first describe failure cases our model. Then, we describe the newly added experiments.

### Failure Cases

#### 1. Failure cases of the detection module.
[This link](https://sites.google.com/view/falcon-iclr/home/failure_detection) provides visualizations for failure cases of the detection module. They have also been included in the appendix.

In general there are two failure modes: miss detection and false positive detection. Both types of detection error may lead to an incorrect object being referenced during program evaluation.
These two types of errors generally occur when objects are highly occluded or in very cluttered scenes.

We found that our program execution module can handle false positive detections to a certain degree. However, since we have pretrained and fixed the detection module, miss detection can not be recovered. Both errors may affect the all stages of training and evaluation. It remains underexplored how to inform the learning of object detection modules based on language information.

#### 2. Failure cases of the semantic parsing module.
[This link](https://sites.google.com/view/falcon-iclr/home/failure_parser) provides visualizations for failure cases of the semantic parser. They have also been included in the appendix. We recognize that there areare two types of failure cases for the semantic parser: syntactic error and semantic error.

In the syntactic error case, the model may build a synthetically wrong program. For example, there is an incorrect number of arguments to an operation. Our implementation handles such kind of error by removing invalid operations, following IEP (Johnson et al. [1]).

In the semantic error case, the model makes a synthetically correct but semantically wrong program. Executing such programs may lead to wrong answers. Currently, there is no mechanisms implemented trying to recover from such errors. In the future, we hope to incorporate methods such as REINFORCE (as in Yi et al. [2] and Mao et al. [3]) to leverage visual information to refine the semantic parser.

#### 3. Failure cases of the concept learning module.
Even if there are no errors in the detection module and the semantic parsing module, our system may still produce wrong answers due to impefect visual grounding of certain concepts. We visualized a few cases in [this link](https://sites.google.com/view/falcon-iclr/home/failure_concept). They have also been included in the appendix.

---

> ### Author Response · Authors · 2021-11-20
> **General Response - Continued**
>
>
> ### Additional Experiments
>
> Per request by the reviewers, we have added two additional abaltion studies.
>
> #### 1. The Effect of the number of base concepts.
> Reviewer 2P5Z poses a question on how the number of base concepts can contribute to the model performance. We design a new split of the CUB dataset derived from 50 training species (130 base concepts), 50 validation species (81 concepts), and 100 test species (155 concepts). In our original experimental setup, we have used 100 training concepts (211 base concepts).
>
> We perform the same fast concept learning experiments on this new split, using our model FALCON-G, a concept-centric baseline NSCL+GNN, and an end-to-end baseline MAC. All concept-centric models use box embeddings spaces. The results are as follows:
>
>
> | # of base concepts | FALCON-G | NSCL+GNN | MAC |
> | -------- | -------- | -------- | -----|
> | 130 | 76.32 | 75.21 | 65.16
> | 211 |81.33 | 78.50 |73.88
>
>
> Since we leverage the relationship between the novel concept and known concepts during the learning of the novel concept, all methods have an accuracy drop when there are fewer base concepts. This demonstrates that transfering knowledge from concepts already learned is helpful. Moreover, our model FALCON-G still has the highest accuracy when the number of base concepts is reduced.
>
> #### 2. The effect of the number of related concepts in supplemental sentences.
> Reviewer 2P5Z poses a question on how the number of supplemental sentences would affect model performance. Since all information captured in the supplemental sentences are the names of related concepts and their relations with the novel concept, we provide an ablation on the effect of the number of related concepts in supplemental sentences.
>
> In the following experiments, we evaluate several models on the fast concept learning tasks on the CUB dataset. For each model, we use supplemental sentences containing 0% (no supp. sentence), 25%, 50%, 75%, and 100% (the setting described in the paper) of all related concepts. Again, we compare our model FALCON-G, a concept-centric baseline NSCL+GNN, and an end-to-end baseline MAC. All concept-centric models use box embeddings spaces. The results are as follows:
>
> |% of all related concepts|FALCON-G|NSCL+GNN |MAC|
> | -----|----|----|---|
> |0|76.37|73.38|73.55|
> |25|80.20|77.16|73.94|
> |50|80.78|76.93|74.13|
> |75|81.20|77.77|74.23|
> |100|81.33|78.50|73.88|
>
>
> [This link](https://sites.google.com/view/falcon-iclr/home/number_of_related) provides figure visualizations for how the model accuracy changes w.r.t. the number of related concepts in the supplemental sentence. We have the following observations:
>
> 1. More related concepts included in the supplemental sentence generally leads to higher accuracy, across all models.
> 2. In both concept-centric models (FALCON-G and NSCL+GNN), the most significant improvement in test accuracy occurs between 0% and 25%. This suggests that these models can benefit from even just an incomplete set of related concept information.
> 3. Our model, FALCON-G, performs consistently the best across all percentages of related concepts.
>
>
> ### Conclusion
> To summarize, we thank the reviewers for their careful feedback and additional suggestions for evaluation, which will make the paper significantly stronger. We look forward to further discussion, and are happy to answer any questions that might arise.
>
> **References**
>
> [1] Justin Johnson, Bharath Hariharan, Laurens van der Maaten, Judy Hoffman, Li Fei-Fei, C. Lawrence Zitnick, and Ross Girshick. Inferring and Executing Programs for Visual Reasoning. In ICCV, 2017.
>
> [2] Kexin Yi, Jiajun Wu, Chuang Gan, Antonio Torralba, Pushmeet Kohli, and Joshua B. Tenenbaum. Neural-Symbolic VQA: Disentangling Reasoning from Vision and Language Understanding. In NeurIPS, 2018.
>
> [3] Jiayuan Mao, Chuang Gan, Pushmeet Kohli, Joshua B. Tenenbaum, and Jiajun Wu. The Neuro-Symbolic Concept Learner: Interpreting Scenes, Words, and Sentences From Natural Supervision. In ICLR, 2019.

---

### Decision · Program_Chairs · 2022-01-20

**Decision:**

Accept (Poster)

**Comment:**

This paper presents a meta learning framework to learn novel visual concepts with few examples. The proposed FALCON model uses an embedding prediction module to infer novel concept embeddings. This is done via paired image and text data as well as supplementary sentences. The resulting systems shows improvements on a series of datasets with synthetic and real images. The reviewers were supportive of this submission and praised the novelty, central ideas and experimental setups.

Concerns included:\
(a) [2P5Z] Justifying the formulations in this paper and situating it with past work -- "Why is this an ecologically valid problem formulation?", "why a meta-learning approach is the best formulation to tackle the problem?", "Why the box embedding space?"\
(b) [W3YC] More details required about the dataset and approach.\
(c) [98FU] Failure patterns

The authors provided detailed responses to these concerns. Concern (a), (b) and (c) were well addressed in the rebuttal and paper, and led to in increase in the reviewers rating.

Given the above, I recommend acceptance. But I do urge the authors to add the details provided in the rebuttal into the main paper. In particular, the concerns/suggestions by reviewer 2P5Z can hugely help in improving the paper and informing the reader.